# Accurate prediction of the optical properties of nanoalloys with both plasmonic and magnetic elements

Vito Coviello [1], Denis Badocco[1], Paolo Pastore[1], Martina Fracchia[2,3], Paolo Ghigna [2,3], Alessandro Martucci [3,4], Daniel Forrer [1,5] ✉ & Vincenzo Amendola [1,3] ✉

The alloying process plays a pivotal role in the development of advanced multifunctional plasmonic materials within the realm of modern nanotechnology. However, accurate in silico predictions are only available for metal clusters of just a few nanometers, while the support of modelling is required to navigate the broad landscape of components, structures and stoichiometry of plasmonic nanoalloys regardless of their size. Here we report on the accurate calculation and conceptual understanding of the optical properties of meta-stable alloys of both plasmonic (Au) and magnetic (Co) elements obtained through a tailored laser synthesis procedure. The model is based on the density functional theory calculation of the dielectric function with the Hubbard-corrected local density approximation, the correction for intrinsic size effects and use of classical electrodynamics. This approach is built to manage critical aspects in modelling of real samples, as spin polarization effects due to magnetic elements, short-range order variability, and size heterogeneity. The method provides accurate results also for other magnetic-plasmonic (Au-Fe) and typical plasmonic (Au-Ag) nanoalloys, thus being available for the investigation of several other nanomaterials waiting for assessment and exploitation in fundamental sectors such as quantum optics, magneto-optics, magneto-plasmonics, metamaterials, chiral catalysis and plasmon-enhanced catalysis.

Plasmonics is the branch of photonics dealing with all those physical phenomena rising when the conduction electrons in a material are collectively excited by an electric field[1,2]. In nanoparticles (NPs) and nanostructures, conduction electrons can be excited also by electromagnetic radiation, leading to surface plasmons and localized surface plasmons (LSP)[1,2]. Over the last decades, plasmonics permeated a wide range of sectors with critical importance for future technological challenges, such as sensing[3,4], photocatalysis[5–7], sunlight conversion[8], metamaterials[9], nanomedicine[10], nonlinear[11] and quantum[12,13] optics. Each of these applications has specific requirements that cannot be simultaneously fulfilled by a single material[14–16]. Hence, one main challenge existing in the field of plasmonics is how to rationally tailor the properties of a material to comply with the restrictive requirements for real-world applications[14–17].

It is meaningful that the vast majority of plasmonic phenomena have been reported in systems composed only of gold and silver[15–19].

[1]Department of Chemical Sciences, Università di Padova, via Marzolo 1, 35131 Padova, Italy. [2]University of Pavia, Department of Chemistry, viale Taramelli 16, 27100 Pavia, Italy. [3]INSTM, National Inter-University Consortium for Materials Science and Technology, Via G. Giusti 9, 50121 Florence, Italy. [4]Department of Industrial Engineering, University of Padova, Via Marzolo 9, 35131 Padova, Italy. [5]CNR – ICMATE, via Marzolo 1, 35131 Padova, Italy. ✉e-mail: daniel.forrer@unipd.it; vincenzo.amendola@unipd.it

Just a small fraction of studies so far considered compounds with lower chemical stability such as Al[20], Ga[14] and Cu[21], or alloys. In particular, the attributes of traditional plasmonic elements (Au, Ag) have been prevalently expanded by alloying with other coinage metals (Cu)[22], noble metals (Pd, Pt)[23–25], d-metals (Fe)[26,27] or sp-metals (Al)[28]. These examples pointed to alloying as a leading strategy to combine different physical and chemical properties at the nanoscale and achieve distinct phenomena with a potentially disruptive impact on the current landscape of photonic technologies[7,15–19]. Nonetheless, the library of the possible permutations is wide and the accurate experimental mapping of a reasonable number of combinations is made cumbersome by several synthetic challenges[15,18,29,30]. These are connected to the different reactivity and stability of the elements included in the nanoalloys, as well as to thermodynamics, which hampers a considerable part of the permutations of plasmonic elements with the other elements in the whole stoichiometric range[15,30]. Besides, the accuracy of current structural investigation methods can reach atomic precision only in highly ordered systems, such as intermetallics[16,29,31]. On the contrary, the short-range order, atomic scale segregation and chemical state of the elements in disordered systems such as solid solutions or heterogeneous NPs are very difficult to be assessed, even with high-resolution (HR) electron microscopy or X-ray diffraction (XRD) methods, and are generally ignored[16,29]. Consequently, the role of the chemical order in the physical and chemical behavior of nanoalloys is still not well understood and the possibility to study its effect through accurate computational approaches would be a major advance in the field. Besides, a nanoalloy cannot be benchmarked, by definition, with the optical properties previously reported in literature or with existing models with limited accuracy, as it would be necessary to confirm alloying, provide information on the effective element distribution in the lattice and consolidate the conceptual understanding of these systems.

In this framework, the availability of first-principles methods that can provide an accurate and reliable description of the optical properties of nanoalloys at a sustainable computational cost is thus of critical importance for the research efforts towards the development of plasmonic nanoalloys[15–17,26,32]. For alloy NPs with a size range above just a few nm[33], the optical properties are accessible with excellent accuracy by the knowledge of the material dielectric function and the use of classical electrodynamics Mie model[17,26,34,35]. The density functional theory (DFT) is the reference computational approach for first-principles calculation of electronic structure and properties like the dielectric function in materials[6,36]. However, the capability to describe the plasmonic nanoalloys relies on the predictive accuracy of the models as well as on their ability to capture the complexity of realistic systems[32,36]. For instance, the dielectric functions of plasmonic alloys of coinage and noble metals, which have full $d$ levels and are non-magnetic, were recently calculated with linear-response time-dependent DFT (TD-DFT) and quantitatively compared with experimental results in terms of plasmon peak position and width, while no information was provided about the accuracy of the absolute extinction coefficient in real samples of plasmonic nanoalloys[17,37]. Therefore, no information can be extracted on the quantitative reliability expected for the prediction of other important plasmonic properties such as, for instance, the local field enhancement, the generation rate of hot carriers, the chiroplasmonic and the magneto-optical response. Moreover, the agreement was good only for the compounds of isovalent elements (Au, Ag, Cu), that have a quasi-linear modulation of the plasmonic response between the pure metals. Conversely, the model had several pitfalls for alloys of non-isovalent coinage and noble metals. In these cases, despite their full $d$ levels, the optical properties are markedly different from the simple average of the properties of the constituting elements[15,17]. Systematic computational studies have been reported also for some alloys of plasmonic elements with $d$- or $sp$-metals[38], but the comparison with experimental results has not been attempted except for the Au-Al system[39].

Another promising approach has been recently exploited for the calculation of the dielectric function in plasmonic alloys of Au and Ag, which is based on the Independent Particle Approximation (IPA) applied on top of the Hubbard-corrected DFT (DFT + U) band structure[40,41]. The Hubbard term can mitigate the delocalization error typical of DFT because it has the effect of localizing strongly correlated electrons, such as those in $d$ bands, whose energy is typically overestimated by DFT[40,41]. This happens through the introduction of a Hubbard parameter U acting selectively on atomic orbitals which, usually, are occupied by strongly-correlated electrons such as those in $d$ bands. The value of U must be optimized, preferably against available experimental data, such as the dielectric function of elemental constituents of the alloy, to match the targeted properties of the system. This can be performed easily because the dielectric functions are available in literature for pure elements and for several alloys. The combination of DFT + U and IPA is computationally less expensive than generally accurate approaches such as the Bethe Salpeter Equation (BSE), while still providing reliable results on alloys of plasmonic elements[40,41]. Orbital-dependent exchange-correlation functionals, such as the GLLBSC one[42], showed promising results for the calculation of dielectric functions in plasmonic alloys[17,32]. The most important feature of the GLLBSC potential is to vary abruptly when the orbital occupation changes by a small amount[43]. This feature recovers the correct behavior of the exchange energy, but is challenging for diagonalization algorithms which become unstable for materials having very narrow $d$-band pseudo-gap, such as materials with magnetic elements[36]. The Hubbard correction, on the contrary, does not introduce particular issues and, through the U parameter, provides the flexibility to adapt the method for the specific task of interest[40,41].

Nevertheless, the DFT + U method was not yet applied to the quantitative prediction and conceptual understanding of the optical properties of real samples of alloy NPs. Especially for nanoalloys with plasmonic and magnetic elements, which are important for catalysis and various quantum optic phenomena[15,18,29], there are no examples where the experimental optical properties have been investigated or interpreted starting from the calculation of the dielectric function. Even for pure noble metals, the experimental measurement of the dielectric function by ellipsometry is difficult, due to its dependence on film density, roughness, purity and measurement range, leading to different results in literature[17,44,45]. This task becomes very challenging in the case of multielement metals subject to oxidation or forming metastable alloys prone to segregation[17,18,46]. Indeed, the advantage of using computed dielectric functions in place of experimental ones is that of overcoming both the experimental challenges related to their measurement and the thermodynamic constraints that hinder the access to optical properties of metastable materials.

Here we consider the prediction and quantitative assessment of the optical properties in alloys of plasmonic (Au) and magnetic (Co) elements. Au is the most chemically and physically stable plasmonic element, biocompatible and renowned for its surface chemistry allowing versatile functionalization[34,47–49]. Besides, Au has interesting catalytic properties for a series of photocatalytic and photo-electrocatalytic reactions of strategic importance for developing a sustainable economy[34,47–49]. Co is an earth-abundant element with the second highest atomic magnetic moment in its pure metallic state (1.72 $\mu_B$), possesses plasmonic properties when not oxidized, and exhibits the catalytic properties of the transition metals of the 4th period[50,51]. Hence, the Au-Co nanoalloy is of interest for quantum plasmonics, magneto-plasmonics, chiral catalysis, responsive optical nanomaterials, sensing and, in general, for benchmarking the research capabilities in this field[15,18,29]. However, little is known about the optical properties of this magnetic-plasmonic nanomaterial, also because the Au-Co solid solution is metastable which, together with the diverse oxidation potentials of the two metals, imply the maximum synthetic challenges for a nanoalloy[29,51–53]. To solve this problem, state-of-the-art

techniques were used for the synthesis by laser ablation in liquid (LAL) and the sorting of colloids of metastable alloys[26,27,53–55]. These difficulties explain why accurate experimental and theoretical dielectric functions are not available for the Au-Co alloys. Overall, the Au-Co NPs of this study constitute one of the most challenging tests for the synthesis and modelling of bimetallic plasmonic nanoalloys, while also being representative of their wide potential and application landscape. Therefore, we employed the DFT + U method alongside with the IPA as the starting point to model and understand the optical properties of the experimental samples. The integrated computational-experimental endeavor reproduced the optical properties of realistic systems such as the colloidal solutions of Au-Co plasmonic nanoalloys with different composition and size distribution, with coefficients of determination ($R^2$) larger than 0.97. The general applicability of the method was also substantiated with another magnetic-plasmonic nanoalloy (Au-Fe) and a typical plasmonic nanoalloy (Au-Ag), as well as by comparison with experimental and calculated bulk dielectric functions. Hence, this study shows that modelling the optical properties of nanoalloys can meet the accuracy level of the experiment, which is the starting point for the mindful development of innovative plasmonic nanoalloys with tailored properties.

## Results

### Synthesis and characterization of Au-Co nanoalloys

Colloidal synthesis is an important way to obtain suitable nanostructures for the many branches of photonics[15,16]. Here, Au-Co alloy NPs are obtained by laser ablation of a bulk alloy target in ethanol (Fig. 1a)[53,56]. Due to the presence of Au atoms, the surface of the laser-generated NPs is ready for conjugation with thiolated hydrophilic polymers (polyethylene glycol, PEG) to gain complete colloidal stability in aqueous solution[27,53], where the nanoalloys were transferred simply by dialysis. The Au-Co nanoalloys have a spherical shape and crystalline structure, as evidenced by the HR transmission electron microscopy (TEM, Fig. 1b). LAL is a highly energetic synthesis process, as required to freeze the matter in a metastable state[15,29], but this implies that the so-obtained Au-Co NPs are polydisperse in size. Hence,

a sedimentation-based separation (SBS) protocol[27,55] in pure water (Fig. 1c) was adopted to harvest three fractions with a much lower polydispersity and an average size of, respectively, 45 ± 18 nm (sample A), 24 ± 6 nm (sample B), and 6 ± 3 nm (sample C). These three samples cover a size range from those subject to intrinsic size effects in plasmonic NPs, relevant when the dimension is comparable to the mean free path of the conduction electrons and due to the surface contribution to the plasmon relaxation rate, to that of extrinsic size effects, relevant in NPs with size not far from the plasmon excitation wavelength and due to electric field inhomogeneity over particles volume (retardation effects)[34,57]. The bidimensional map of the elemental composition in the three samples was collected by scanning TEM (STEM) energy dispersive X-ray (EDX) spectroscopy (Fig. 1d), confirming with nanometric resolution the homogeneous distribution of Au (M-edge) and Co (K-edge) in all the alloy NPs.

Despite their different size distribution (Fig. 2a), all Au-Co samples analyzed with XRD exhibit the same reflections of the face-centered cubic (FCC) lattice that could be found in pure Au (Fig. 2b), without any contribution due to other phases containing Au or Co. The Rietveld refinements of the XRD patterns indicate a cell parameter of 4.0038 ± 0.0002 Å (sample A), 4.0043 ± 0.0007 Å (sample B) and 4.0328 ± 0.0006 Å (sample C). These cell parameters are contracted compared to pure Au (4.079 Å), which is consistent with the formation of a solid solution where Co, which has a smaller atomic size (1.25 Å) than Au (1.44 Å)[51], is a random substitutional impurity in the gold lattice[49,51,52]. A bimetallic compound forming a solid solution with the same lattice of the pure metal is a textbook case for the application of Vegard's law on the linear dependence of the cell parameter to the alloy composition[58]. The use of Vegard's law for alloys is justified because STEM-EDX allowed excluding the formation of core-shell structures (see Fig. 1d and Suppl. Note 1), which is possible in NPs of immiscible elements[51,59,60]. This allowed the evaluation of Co content reported in Fig. 2c (hollow squares). The composition of the NPs was further assessed by STEM-EDX on small groups of NPs (hollow triangles) and by Inductively coupled plasma mass spectroscopy (ICP-MS, hollow circles), resulting in an average composition of Au(76)Co(24)

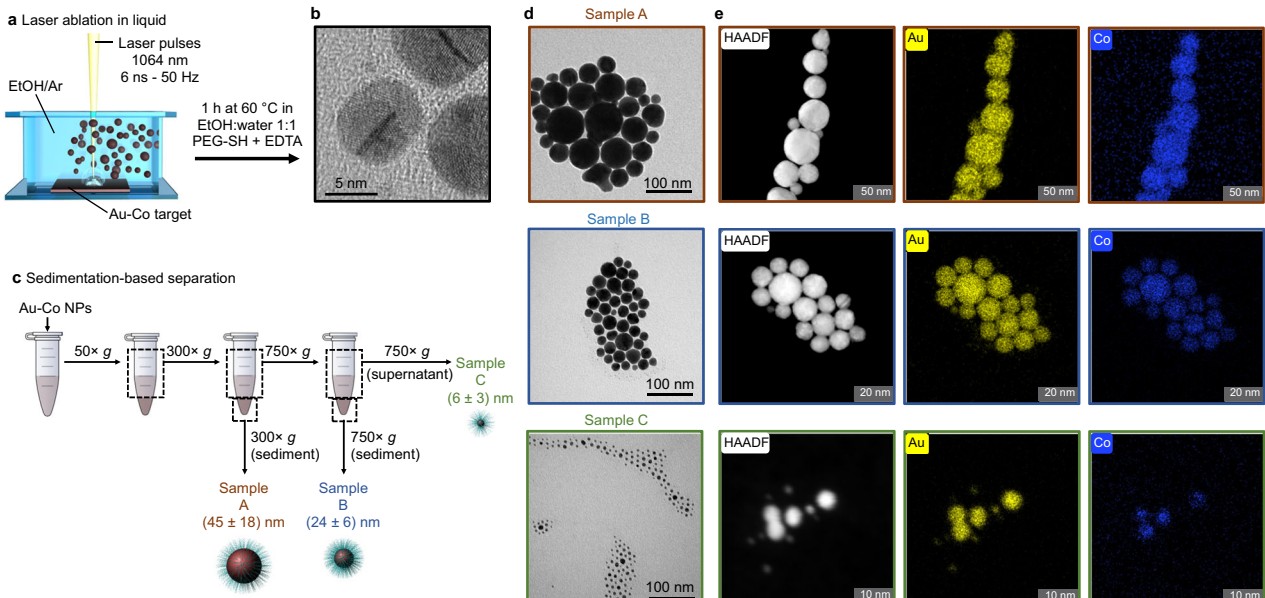

**Fig. 1 | Synthesis and TEM characterization of Au-Co NPs. a** Schematic depiction of the LAL synthetic protocol to obtain metastable Au-Co alloy NPs: ns laser pulses are focused on a bulk Au-Co target to generate a colloid of nanoalloys in ethanol. **b** HRTEM image of the Au-Co NPs. **c** Schematic depiction of the SBS protocol to isolate different size fractions from the pristine sample. In SBS, the PEG-coated Au-Co NPs in water are subjected to centrifugal fields with increasing centrifugal force to separate NPs with different average size. **d**, **e** TEM (**d**) and STEM-EDX (**e**) bidimensional images of the three Au-Co samples, showing a homogeneous distribution of the two elements in the NPs (Au in yellow, Co in blue).

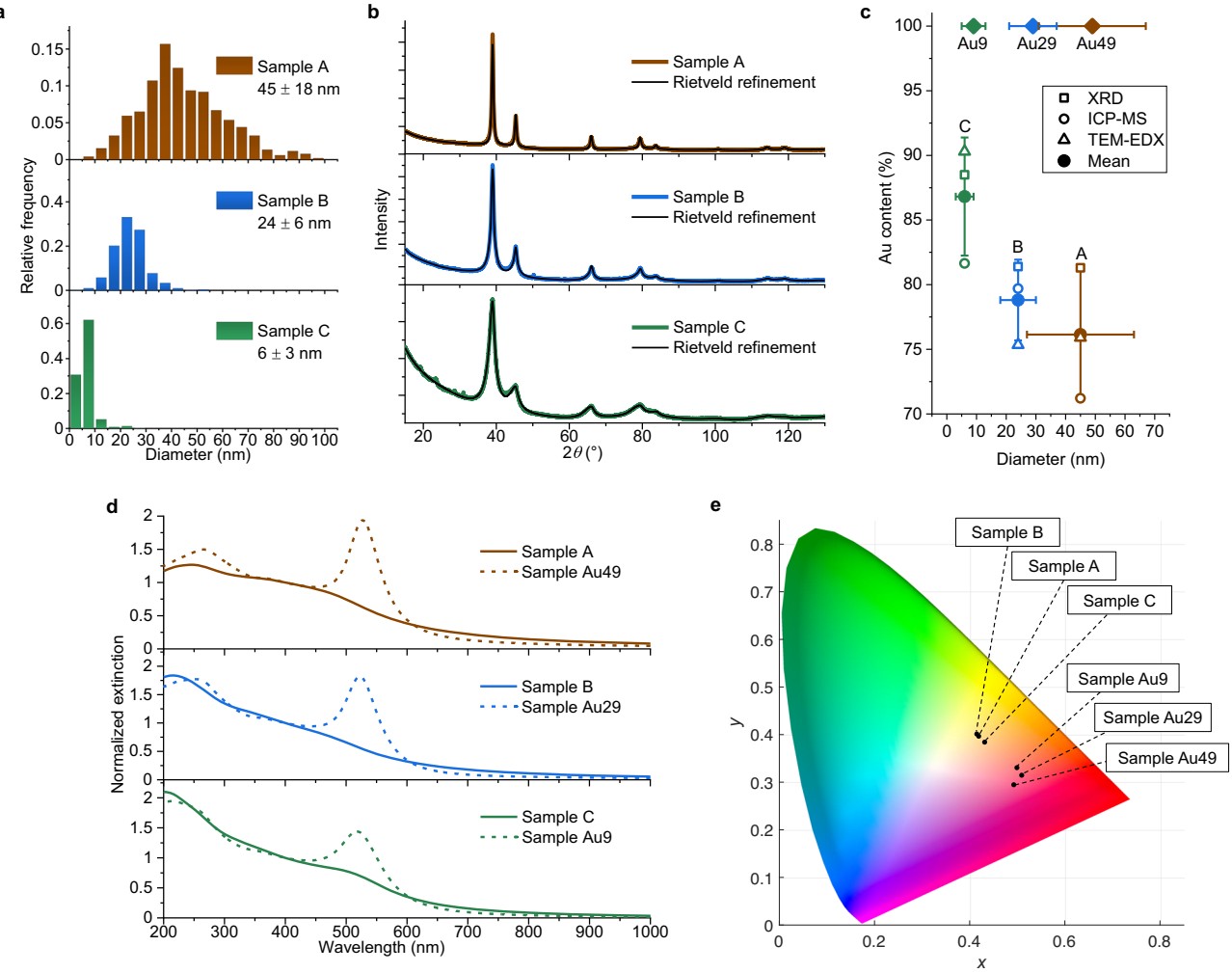

**Fig. 2 | Optical and structural characterization of Au-Co NPs. a** TEM-measured size histogram of the Au-Co NPs samples (sample A (brown): 45 ± 18 nm; sample B (blue): 24 ± 6 nm; sample C (green): 6 ± 3 nm). Statistics considered >500 NPs for each sample. **b** XRD pattern and Rietveld refinement of the Au-Co NPs samples. Colors in the graphs are: brown (sample A); blue (sample B); green (sample C). **c** Plot of composition and size of the Au-Co NPs samples (brown: sample A; blue: sample B; green: sample C) and pure Au NPs samples with similar size used for comparison (brown: sample Au49; blue: sample A29; green: sample Au9; see Methods). The diameter was determined from TEM image analysis and horizontal error bars indicate the size standard deviation from TEM histograms. Vertical error bars indicate the semi-dispersion over the mean composition assessed with the three techniques indicated in the legend. **d** Optical absorption spectra of the Au-Co NPs samples (brown: sample A; blue: sample B; green: sample C) and pure Au NPs samples with comparable size (brown dashed line: sample Au49; blue dashed line: sample A29; green dashed line: sample Au9). The data was normalized to the extinction at 400 nm. **e** Position of the six NP samples in the CIE diagram, showing the different locations due to the different plasmon responses. Source data are provided as a Source Data file.

for sample A, Au(79)Co(21) for sample B and Au(87)Co(13) for sample C (Fig. 2c). The average composition was confirmed further by additional STEM-EDX quantitative analysis performed on several single NPs of each sample (Supplementary Table 1).

The optical properties of the Au-Co nanoalloys in water can be assessed with UV-vis spectroscopy (Fig. 2d) and are much different than those of pure Au NPs with comparable size and size distribution (sample Au49: 49 ± 18 nm, sample Au29: 29 ± 8 nm, sample Au9: 9 ± 4 nm). In particular, the spectra of pure Au samples are known for the LSP band at 520 nm, over imposed on the edge of *d-sp* interband (IB) transitions which dominate the optical density in the near UV[34]. In the Au-Co samples, only the absorption edge typical of the *d-sp* IB transitions is well recognizable. Sample A exhibits a broad band in the 475–500 nm range, which could be attributed to a blue-shifted LSP or to IB transitions. The LSP is not well identifiable in sample B, while a weak plasmon absorption centered at 505 nm is observed in sample C (i.e. the sample richest in Au). Noteworthy, the optical density of all the Au-Co samples never reaches zero in the visible range, suggesting that a broad LSP band is present in all cases. The spectral differences are

qualitatively appreciable also from the colors calculated from the optical extinction spectra of the samples (Fig. 2d), which are red-brownish for the Au-Co nanoalloys and purple-red for the reference Au NPs, corresponding to different positions in the International Commission on Illumination (CIE) diagram (inset of Fig. 2e).

## Modelling of the optical properties

The agreement of the experimental optical spectra with those expected for a colloid of Au-Co nanoalloys in water can be verified using the Mie model for spherical particles and providing as input the experimental size histogram measured with TEM, the dielectric function of water for the particles' environment, and the alloy dielectric function for the specific stoichiometry of each sample[34,57]. Since no experimental dielectric functions are available in literature for the Au-Co alloys, instead of resorting to inaccurate methods such as the weight of single metal dielectric functions, the DFT + U method coupled with IPA was used to calculate these functions[40,41]. The overall procedure (Fig. 3a–c) starts with the parametrization of U for Au 5*d* and Co 3*d* states versus the experimental dielectric function of pure Au and Co

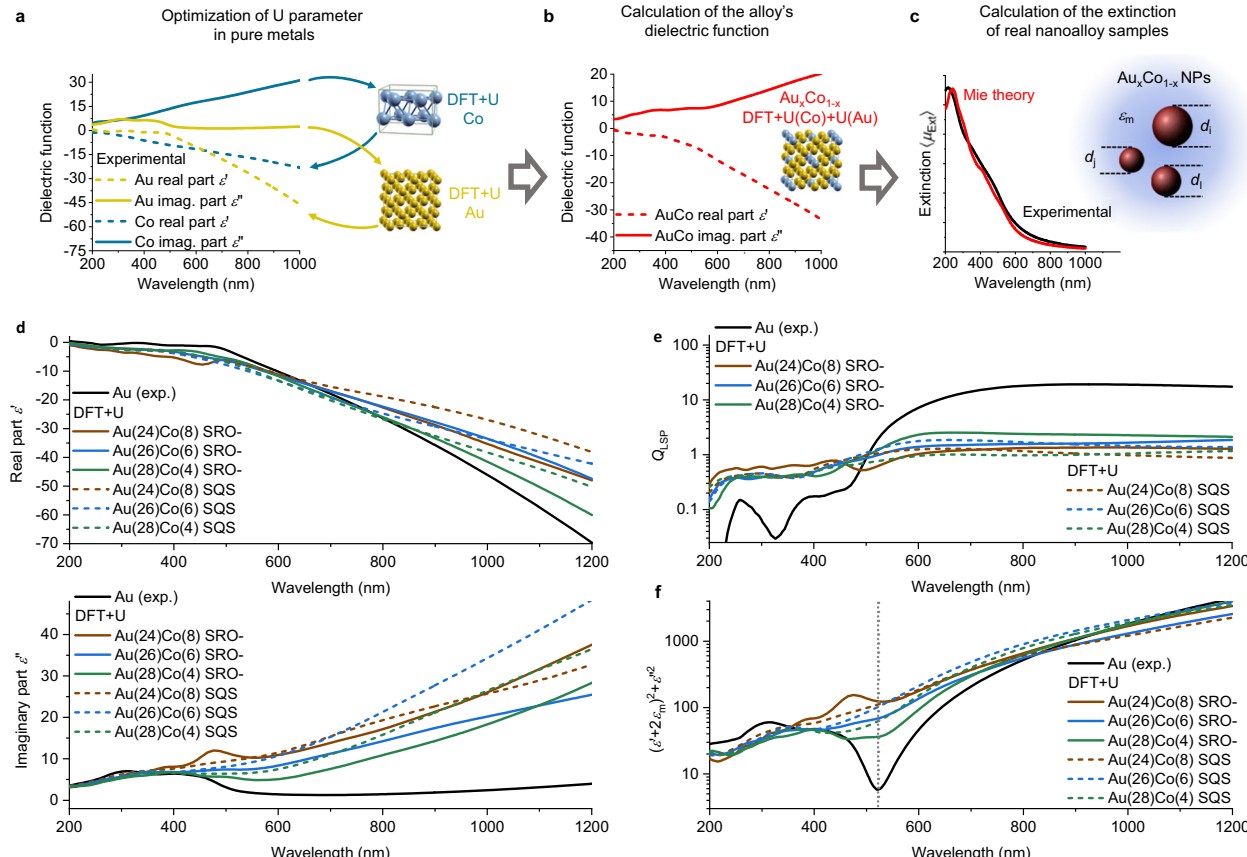

**Fig. 3 | Modelling approach and calculated dielectric functions for the Au-Co nanoalloys. a–c** Schematic depiction of the procedure for modelling the optical properties of Au-Co alloy samples: (**a**) optimization of U by modelling the experimental dielectric function of pure Au and Co (yellow: Au; blue: Co); **b** DFT + U calculation of the dielectric functions of the Au-Co alloys with the same composition of the experimental samples; **c** Mie theory calculation (red) of the extinction of the colloid of Au-Co alloy NPs (black) using the computed dielectric functions for the alloy, the experimental dielectric function of the solvent and the size

distribution measured by electron microscopy. **d** $\varepsilon'$ and $\varepsilon''$ obtained by DFT + U calculations with SRO- or SQS models, compared to the experimental value of pure Au (brown: Au(24)Co(8); blue: Au(26)Co(6); green: Au(28)Co(4); black: Au). **e, f** Plot of $Q_{LSP}$ (**e**) and $(\varepsilon'+2\varepsilon_m)^2 + \varepsilon''^2$ (complete Fröhlich condition in a nanosphere, **f**) using the calculated dielectric functions for the Au-Co alloy and the experimental value for Au. Color in the graphs are brown (Au(24)Co(8)), blue (Au(26)Co(6)), green (Au(28)Co(4)), black (Au). Source data are provided as a Source Data file.

reported in literature (see Supplementary Note 2)[44,61]. Then, the same values of U were transferred to Au-Co alloys with compositions equivalent to the experimental samples. This is legitimate because U changes with the element oxidation state but it remains almost unchanged when going from pure metals to their alloys[41].

The dielectric functions were used in the Mie model to calculate the mass extinction coefficient ($\mu_{ext}$). Since the optical properties of NPs are size dependent, the accuracy of the Mie model was maximized by using the TEM-measured size ($d_i$) histogram of each Au-Co NPs sample as input (i.e. the Mie calculation was performed for each size bin of the size histogram and then weighted for the relative frequency). Besides, at each size bin in the TEM histogram, the extrinsic size effects were accounted for by setting the multipolarity order to 4 and the dielectric function was corrected for the intrinsic size effects with the well-established procedure proposed by Kreibig and based on the Drude-Lorentz model[34,57].

The $\mu_{ext}$ is representative of the plasmonic properties of nanoscale objects, which are the focus of this study. The experimental UV-vis spectra of a colloid of NPs collected with different optical absorption spectrometers are comparable within a photometric accuracy of less than 0.002 in absorbance, and the mass of NPs in the sample can be measured accurately by ICP-MS using certified standards and procedures, therefore $\mu_{ext}$ is an accurate parameter for benchmarking the predictions of our model. The UV-vis and ICP-MS analysis, associated to STEM-EDX, allow a more direct assessment of nanoalloys structure,

morphology and composition compared to thin films used for ellipsometry, where also surface roughness and density should be accounted with cross sectional resolution[44,45].

DFT calculations were performed on two sets of alloys models, namely cells with short-range order (SRO-) obtained by "diluting" Co atoms into next neighbors (NN) made only of Au atoms, and special quasi-random structures (SQS) representative of random solid solutions[62]. This difference can be quantified with the adimensional Warren-Cowley parameter (WC) for the NN shell of Co atoms, where a negative value corresponds to short-range order (as in SRO- models), a nearly zero value to random alloys (as in SQS models), and a positive value to clustering and segregation[63,64]. Alloy models are 2 x 2 x 2 FCC supercells containing 32 atomic sites and with three compositions (Au(28)Co(4), Au(26)Co(6) and Au(24)Co(8)) equivalent to the experimental samples.

The real ($\varepsilon'$) and imaginary ($\varepsilon''$) components of the dielectric functions obtained with the SRO- cell significantly differ from the experimental ones for pure Au (Fig. 3d). This is indicative of the change in the electronic structure of the alloy, compared to pure gold, as expected from the experimental UV-vis spectra and previous studies on Au nanoalloys with transition metals[26]. In particular, the imaginary component $\varepsilon''$ of the alloy is sensibly increased at wavelengths longer than 400 nm, with a monotonous increment from the visible to the near infrared. This is accompanied by a moderate decrease in the absolute value of $\varepsilon'$ in the same spectral region. On the contrary, the

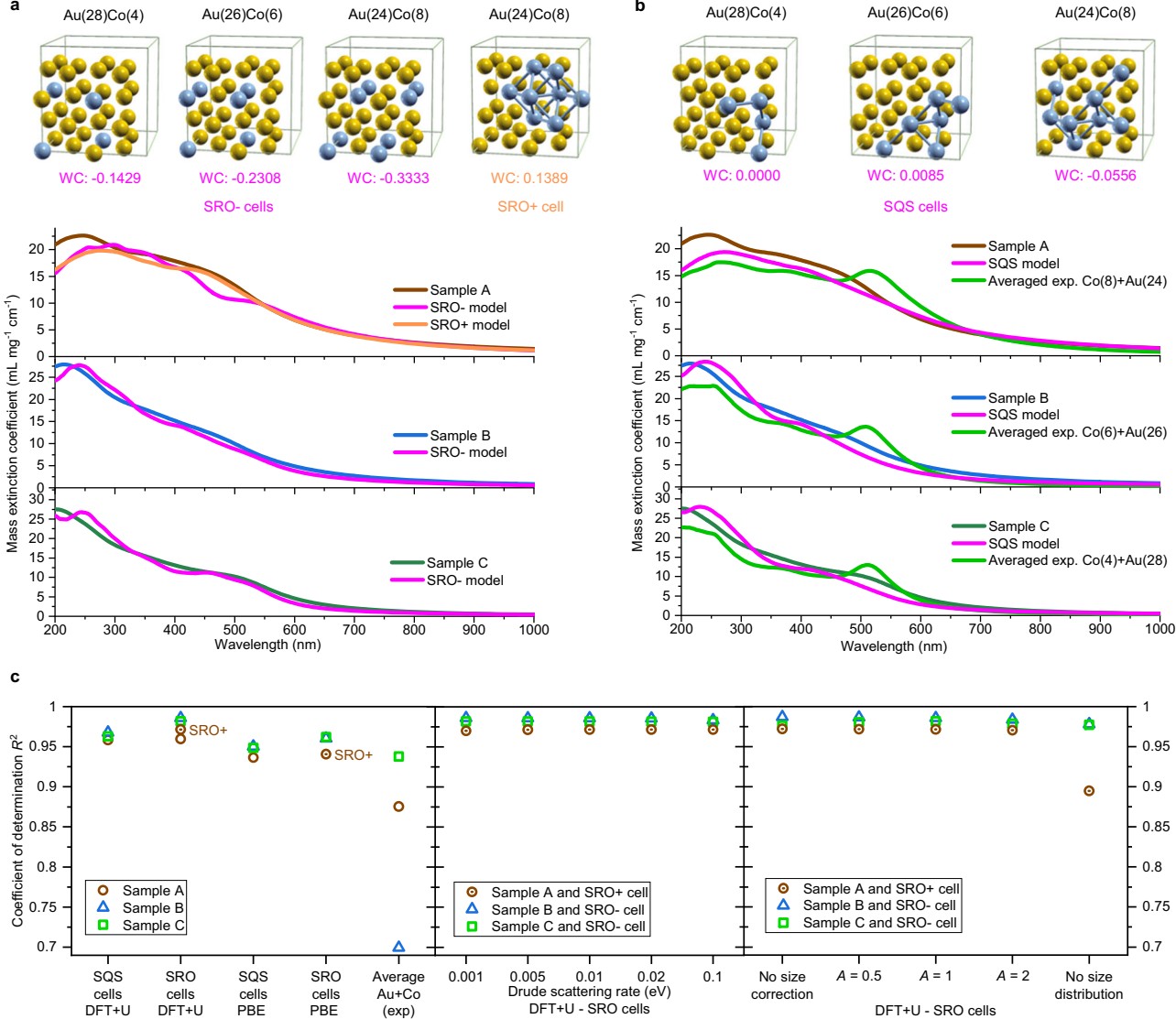

**Fig. 4 | Comparison of experimental and calculated mass extinction coefficient (in mL mg⁻¹ cm⁻¹) for the three samples of Au-Co alloy NPs in water. a** DFT calculations with the SRO− (magenta) and SRO+ (orange) cell model. **b** DFT calculations with the SQS cell model (magenta) or with a composition-weighted average of the dielectric functions of pure Au and Co (light green). The WC parameters which quantify the short-range ordering in each cell are also reported in (**a**, **b**). **c** $R^2$ quantifying the accuracy of calculated data to experimental ones for the various parameters and models considered in the study (for details see text). Colors in the graphs are: brown (sample A); blue (sample B); green (sample C). Source data are provided as a Source Data file.

dielectric function of the alloy is similar to Au below 300 nm, that is the region dominated by $d$-$sp$ IB transitions in gold atoms[17,26,34,35,37]. This trend is even more accentuated in the dielectric function calculated with the SQS (dashed lines in Fig. 3d), suggesting that the atomic arrangement is relevant for the prediction of the optical properties.

The plasmonic performances of spherical NPs, which can be estimated in the quasistatic regime with a dimensionless quality factor ($Q_{LSP}$) given by the ratio of $-\varepsilon'$ to $\varepsilon''$[1,14], is much lower for the alloy (Fig. 3e), in agreement with the absence of a sharp LSP peak in the experimental spectra of Fig. 2d. The increase of $\varepsilon''$ and the slight decrease of $\varepsilon'$ are also responsible for the blue shift of the weak LSP in Au-Co NPs, whose position for spherical NPs in dipolar approximation is given by the wavelength which minimizes $(\varepsilon'+2\varepsilon_m)^2 + \varepsilon''^2$ (Fröhlich condition in a nanosphere in a medium with dielectric function $\varepsilon_m$, Fig. 3f)[15,34,57]. In case of the alloy with the composition of sample C, which has an LSP in proximity of 500 nm (Fig. 2d), the Fröhlich condition is also satisfied in the same spectral range (Fig. 3f), confirming the plasmonic nature of the weak band in that sample. The same band

is not found in sample B (Fig. 2d), richer in Co, and indeed the minimum at the Fröhlich condition is barely detectable.

The ability of the calculated dielectric functions to reproduce the optical properties of real samples can be assessed by computing the $\mu_{ext}$ of the Au-Co samples with the Mie model using either the SRO− (Fig. 4a) or the SQS (Fig. 4b) cells and comparing them to the $\mu_{ext}$ quantified by ICP-MS. The agreement between the experimental and the calculated curves in Fig. 4a is qualitatively well evident, especially for the SRO− models, including the blue shift of the LSP.

The $R^2$ (Fig. 4c) was used to quantify the agreement between calculations and experiments among different samples and models. The $R^2$ indicates the good agreement between the SRO- calculations and the experimental curves in all three samples, meaning that the procedure worked well despite the differences in the average size, size distribution and stoichiometry of the alloy NPs.

The optical properties obtained from the SQS model also capture the general trend of the experiment, although with lower $R^2$ compared to the SRO− model. The extinction coefficients calculated with the SQS

model for samples B and C have a lower optical density in the visible range, which transforms into an overestimation in the near UV range. Besides, the SQS simulation for the sample C has an absorption band at 425 nm, which is blue-shifted compared to the experiment and the simulation with the SRO− cell. This is quantified by the lower $R^2$ for the samples B and C.

However, the $R^2$ of the SRO− model for sample A is lower than in samples B and C, resulting in a comparable value with the SQS model. Indeed, the sample A has the largest spread between the Co content estimated from the lattice parameter (19 at%) and that obtained from the ICP-MS analysis (29 at%), suggesting a positive deviation from the Vegard's law which is typically observed in alloys with short-range elements segregation[65,66]. This may be indicative of the faster cooling rate of smaller NPs generated during LAL, which has been shown to favor homogeneous mixing instead of phase segregation in immiscible alloys[59,60]. To gather more experimental confirmation about the short-range order of the samples, the extended X-ray absorption fine structure (EXAFS) at the Au $L_3$-edge was measured and analyzed (see Suppl. Note 3). Since this measurement provides a specific insight into the structural environment of Au atoms, it was possible to calculate the interatomic distance of Au nearest neighbor shell from the fitting of the EXAFS signal. The results (summarized in Supplementary Fig. 7) clearly show a deviation of the gold atoms interatomic distance in sample A, compared to samples B and C, versus the experimentally measured content of Co, which substantiate the short-order segregation in the nanoalloy. Hence, a further model of the Au(24)Co(8) was adopted, which contemplates a short-range Co segregation arrangement (SRO+, see Fig. 4a and Supplementary Note 4), i.e. a positive WC parameter. As shown by the plot of the $\mu_{ext}$ and $R^2$ (Fig. 4b, c), the SRO+ model is the best at reproducing the experimental result for sample A. In this way, the quantitative comparison of the calculated and experimental optical properties of the nanoalloys provided insight on their "ultrastructure", namely the structural features not directly identifiable with standard characterization approaches such as electron microscopy. This is useful for guiding further investigations with complex and expensive equipment which is not easily accessible. These findings provide also further evidence to the previous theoretical indications that the short range order in nanoalloys has a measurable effect on the optical properties, and modelling efforts should be tailored to the actual atomic arrangement of the experimental system under examination[17,41].

Due to the lack of other studies about the quantitative agreement between calculated and experimental extinction coefficients in real samples of magnetic-plasmonic or plasmonic nanoalloys with sizes of several nm, the goodness of $R^2$ (0.97140, 0.98571 and 0.98166 for samples A, B and C) can be appreciated by comparison with the values of 0.94707 (Au49), 0.98718 (Au29) and 0.98324 (Au9), which were achieved for the experimental samples of Au NPs shown in Fig. 2d, using one of the best experimental gold dielectric functions[44] and applying the Mie model to the TEM measured size histograms of the Au NPs (as done for the Au-Co NPs).

In all samples and for both the SRO-, SRO+ and SQS models, the main contribution to the decrease of $R^2$ comes from the UV region, indicating that the ability to predict *d-sp* IB transitions can still be improved in the DFT + U method and represents the field for future developments of this approach. However, the benefits of the calculated dielectric function are well evident in comparison to the optical properties calculated from the average of the experimental dielectric functions of pure Au and Co (green lines in Fig. 4b). In this case, LSP peaks are present due to the Au component, contrary to the experimental spectra, explaining the lower $R^2$ values (Fig. 4c). This is a demonstration that linear averaging of dielectric functions of pure metals is an unreliable procedure for modelling the optical properties of nanoalloys[17,26,27,32].

After assessing that the short-range order is relevant for the accuracy of the calculations, also the effect of the functional type

(Fig. 4c and Supplementary Note 5), the Drude scattering rate, the correction for intrinsic size effects and the $A$ parameter for the size correction were quantified (Fig. 4c). The results confirm that the DFT + U method is advantageous for the accuracy of the calculations, while the Drude scattering rate and the $A$ parameter only affect the 4th or 3rd decimal place of the $R^2$, thus the standard values adopted in literature[17,26,34,41,57] are the appropriate choice in absence of information guiding to the optimization for each specific nanoalloy. The correction for intrinsic size effect accounts for a change on the 3rd decimal place of the $R^2$ in samples A and B, where the plasmon resonance is significantly damped and the NPs size is larger, but it is on the 2nd decimal place of the $R^2$ in sample C, where the plasmon resonance is more intense because of the lower amount of Co in the alloy and the smaller size of the NPs. In fact, the role of the $A$ parameter is expected to be more relevant in nanoalloys with a more intense plasmon band. However, when considering the average size instead of the size distribution, the agreement with the experiment decreases systematically, in particular for the sample with the largest size.

## Modelling of Au-Fe and Au-Ag nanoalloys

The general applicability of the method was further validated with other magnetic-plasmonic and standard plasmonic alloys. First, the predictive ability was tested against the dielectric functions reported in literature for some Au-Fe, Au-Ag and Au-Cu alloys, as well as those experimentally measured in this study (Supplementary Note 6). To keep the approach equal to the Au-Co case and as general as possible, the procedure started with the identification of the appropriate U parameters from the dielectric function of the pure elements, which are found in literature with good accuracy (Supplementary Note 2). In principle, U could be best obtained from the alloy dielectric functions, when these are available and accurate, but ellipsometry measurements are not trivial in case of alloys which are metastable or contain elements prone to oxidation[17,44,45]. This is apparent from the variable agreement between our measurements and the literature values for alloys (Supplementary Fig. 10), even for single elements (Supplementary Fig. 4). Nonetheless, the dielectric functions calculated with the DFT + U for the magnetic-plasmonic and the plasmonic alloys always exhibited a fair agreement with the literature values in the whole spectral range from 200 to 1000 nm (Supplementary Fig. 10).

As it is the focus of this study, the optical properties of real nanoalloy colloidal samples of another magnetic-plasmonic system (Au-Fe) and of a typical plasmonic system (Au-Ag) were considered (Fig. 5). Also in this case, the quality of our method is evident from the $R^2$, which are 0.96129 for sample D (Au(80)Fe(20) NPs), 0.96978 for sample E (Au(83)Fe(17) NPs); 0.97738 for sample F (Au(45)Ag(55) NPs) and 0.93222 for sample G (Au(45)Ag(55) NPs). Note that standard calculation parameters ($A$ term = 1, Drude scattering rate = 0.01 eV) and generic SQS cells were used, without any optimization for each specific sample, which could lead to higher $R^2$. Despite this, the dielectric functions calculated with the DFT + U allowed the prediction of the optical properties of real nanoalloys with higher accuracy than the literature permittivity from experiments[46,67] and TD-DFT GLLBSC calculations[17], as well as than the PBE functional, which failed again in reproducing the plasmonic component of the optical extinction in the nanoalloys (Fig. 5).

## Discussion

Innovative plasmonic nanoalloys are structurally complex, difficult to synthesize and lack a well-identified modelling methodology which is computationally feasible also for NPs with sizes of several or tens of nm[68,69]. This prevents the accurate understanding of the optical properties and their optimization toward specific applications. For instance, the magnetic-plasmonic Au-Co system exhibits an almost complete immiscibility gap in the thermodynamic phase diagram at room temperature[51,52]. According to the Hume-Rothery rules for solid

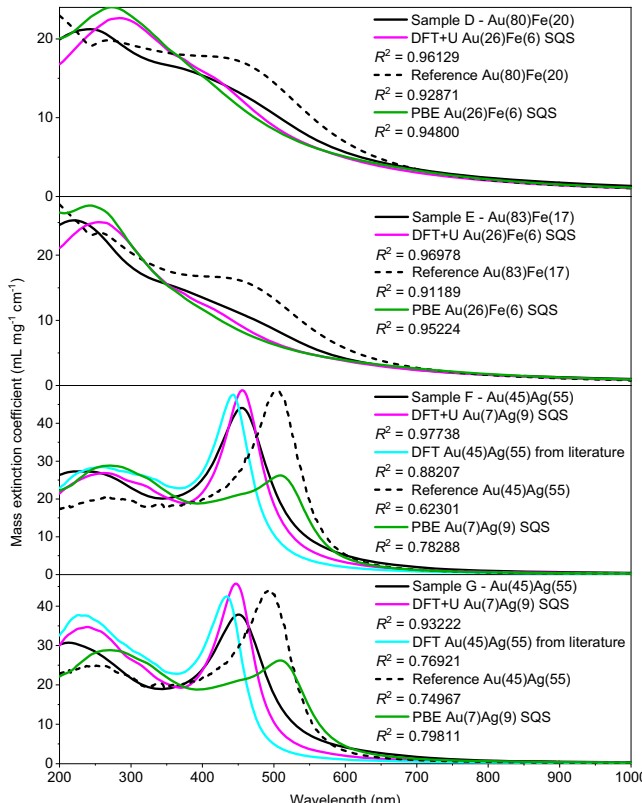

**Fig. 5 | Au-Fe and Au-Ag optical properties.** Comparison of experimental (continuous black line) and calculated (magenta line) mass extinction coefficients (in mL mg$^{-1}$ cm$^{-1}$) for colloidal samples of nanoalloys made of another magnetic-plasmonic system (Au-Fe) and of a typical plasmonic system (Au-Ag). The calculations obtained with the dielectric functions available in literature from experimental measurements (refs. 46,67, black dashed line) and TD-DFT GLLBSC calculations (ref. 17, cyan line), and results with the PBE functional (green line) are also reported. The agreement with the real samples of nanoalloys is evaluated quantitatively with the $R^2$. Source data are provided as a Source Data file.

solutions[15], the differences in atomic radii, crystalline phase of pure elements, electronegativity and number of valence electrons between Au and Co all are too large for alloying. The predictions of the Hume-Rothery rules are confirmed by DFT calculations, because the mixing enthalpy ($\Delta H_{mix}$) of the alloy increases continuously with the Co content in the FCC Au lattice (Fig. 6a). Nonetheless, a highly energetic synthetic method followed by the rapid quenching of atomic kinetic energy, such as LAL, can freeze the two elements in a crystalline solid solution that is a relative thermodynamic minimum[15,29,70]. Despite the difference in atomic volume, the large spatial extension of Au $d$-orbitals permits a good hybridization with $d$-orbitals of transition metal atoms[71], thus contributing to the depth of the local minimum of energy represented by the alloy. In effect, the lattice parameter of the Au-Co NPs samples, evaluated with the Rietveld refinement of the experimental XRD patterns, indicates a reduction compared to pure Au, which is a clear indication of the inclusion of Co in the FCC lattice as a random substitutional dopant[49,51,52]. This contraction is almost linear in the FCC cell volume according to DFT calculations (Fig. 6b), reaching −6.7% (−5.4% experimentally) already at 20 at% of Co.

Although the crystalline structure of the alloy is the same as in pure gold, the presence of a transition metal with partially empty $d$ levels like Co as a substitutional impurity has significant effects on the electronic structure. The electronic density of states (DOS, Fig. 6c) puts evidence on this phenomenon, because additional states with Co $d$ levels character, called virtually bound states (VBS)[1,35,72], appears around the Fermi energy ($E_F$). These are typical of alloys between noble

metals (full $d$ levels) and not-noble transition metals, and enable a wide spectrum of low-frequency single-electron IB (LFIB) transitions[1,35,72]. The VBS are responsible for the increase of $\varepsilon''$ in the visible range, meaning that the formation of electron-hole couples by LFIB transitions is an efficient channel for the relaxation of the plasmon excitation[1,26,35]. The effect is the significant decrease of the $Q_{LSP}$ in Au-Co alloys, corresponding to weak or unidentifiable LSP in samples A, B and C in Fig. 2d. The changes of $\varepsilon''$ and $\varepsilon'$ also imply the blue shift of the LSP peak in spherical Au-Co NPs, according to the complete Fröhlich condition.

In the alloy, the occupied Co $d$ levels contribute to the DOS also in the region located ca. 2 eV below $E_F$, where the occupied Au $d$ levels are found (Fig. 6c and Suppl. Note 7). This is practically appreciable by the change of optical density in the near UV region, which is not reproduced by using the linear average of the dielectric functions for pure Au and Co (as shown in Fig. 4b). Another effect is the upshift of the $d$ band edges towards the $E_F$ and the consequent spectral red-shift of the $d$-$sp$ IB transitions edge compared to pure Au (Fig. 6c). The $d$-$sp$ IB transitions in pure gold are known to be responsible for the lower plasmonic performances of Au compared to Ag, which is the best plasmonic noble metal because has the largest spread between $d$-$sp$ IB transitions edge and LSP energy[1,34,35]. In the time-dependent calculations approaches, this conflict between IB transitions and LSP appears as the effect of the polarization of $d$ levels at the same photon wavelengths at which the plasmon resonates, with the effect of screening the excitation of free electrons in the metal and decreasing the LSP intensity[73–75]. Hence, the red-shift of $d$-$sp$ IB transitions further contributes to the decrease of $Q_{LSP}$ in the Au-Co alloys.

On the other hand, the DFT calculations indicate the increase of the free electron density in the alloy. This is due to two simultaneous effects[26], namely lattice contraction (Fig. 6b) and the larger number of $sp$ electrons in Co (2) compared to Au (1). The $E_F$, which scales with the free electron density, is thus upshifted towards the vacuum level of up to +0.4 eV at Co 25 at% (Fig. 6d). This is supposed to correspond to the decrease of the alloy work function at its interface with the surrounding. A lower work function is appealing and relevant for the exploitation in plasmon-enhanced catalysis and hot-carriers injection in nearby substrates such as molecular adsorbates or semiconductors[5,7,15,76]. The increase of free electron density in the alloy is evident also from the map of the electron density chart $\Delta n(r)$ (Fig. 6e). The plot of $\Delta n(r)$ indicates that the excess charge is effectively delocalized throughout the metal lattice and can be attributed to an actual increase in conduction electron density. Because the plasma frequency scales with the free electron density[77], the increase of $\Delta n(r)$ in the $sp$ bands of the alloy contributes to the blue shift of the LSP towards the $d$-$sp$ IB transition energy[18,26,35], concurring to the decrease of $Q_{LSP}$. The three main effects observed on LSP when Au is alloyed with Co ($d$ band edge upshift, appearance of VBS and increase of $sp$ band electron density) are summarized in Fig. 7.

In summary, the optical properties of colloidal samples of Au-Co alloy NPs with various average sizes, size distribution and stoichiometry were understood and accurately predicted. To this purpose, a procedure was developed based on the first-principles calculation of the dielectric function in alloys of plasmonic and magnetic elements, joined to the classical electrodynamics Mie model and state-of-the-art protocols for the synthesis of metastable colloidal nanoalloys. The spectral features, such as the blue shift and quenching of the LSP band of the alloy compared to pure Au, were predicted and related to the appearance of LFIB, the red shift of $d$-$sp$ IB transitions from Au atoms and the increase of conduction electron density. The accuracy of the computed dielectric function and resulting extinction coefficient is sensitive to the chemical order of the cell model, with the optimal outcomes found for short range order cells with a WC parameter adjusted to the experimental evidence. In the core of this procedure, there are the DFT + U calculations of the dielectric functions after

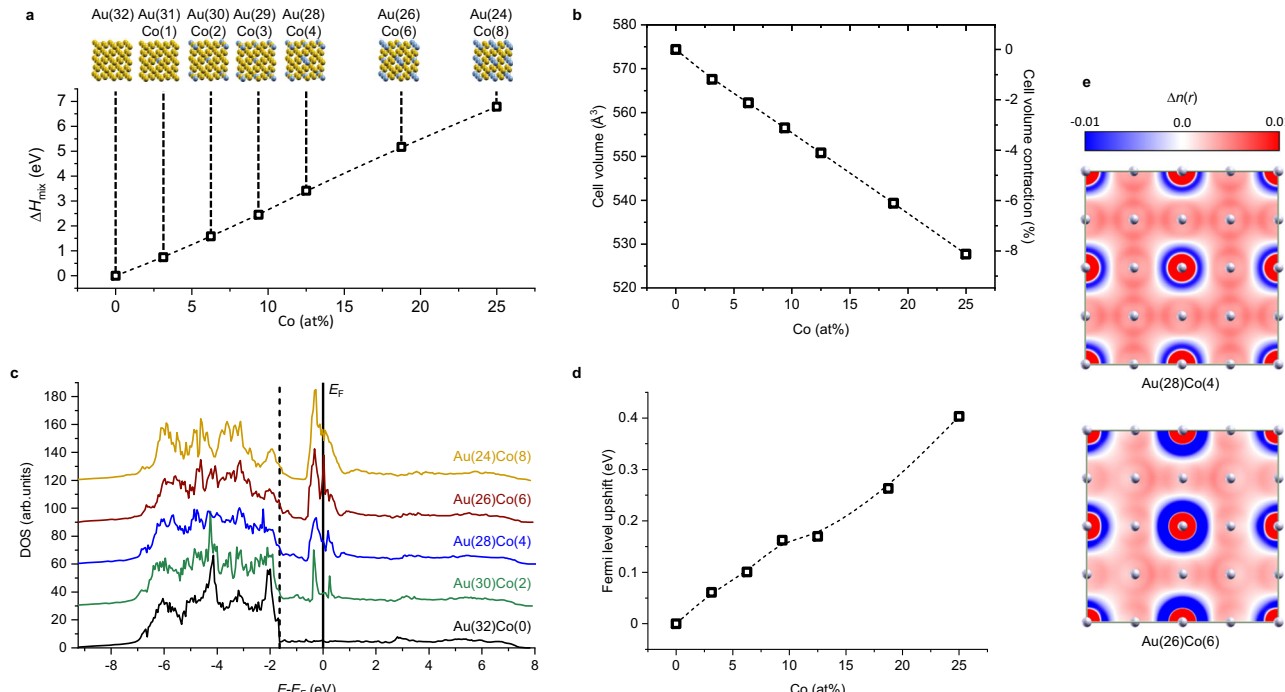

**Fig. 6 | DFT results for Au(32-x)Co(x) cells with x between 0 and 8 (25 at%).**
**a** Mixing enthalpy and models of cubic cells used in DFT calculations (Au atoms in yellow, Co atoms in cyan). The dashed line is a b-spline interpolation of data. **b** Cell volume (in Å or normalized to pure Au value). The dashed line is a b-spline interpolation of data. **c** DOS plot showing the appearance of Co *d* band levels around the Fermi energy ($E_F$). The black dashed line is located at the onset of *d* bands edge in pure Au and emphasizes their upshift with increasing Co content. **d** $E_F$ upshift versus alloy composition. Colors in the graphs are: black (Au(32)Co(0)), green

(Au(30)Co(2)), blue (Au(28)Co(4)), brown (Au(26)Co(6)), light brown (Au(24) Co(8)). The dashed line is a b-spline interpolation of data. **e** Contour plot of the electronic density difference between Au(28)Co(4) or Au(26)Co(6) and pure Au bulk models showing the presence of excess electron density delocalized over the entire structure. FCC sites are indicated as white spheres. Note that a large oscillation of $\Delta n(r)$ appears at sites where Au was substituted with Co, due to the way $\Delta n(r)$ is defined (see Methods). Source data are provided as a Source Data file.

optimization of the U parameters from the experimental data of pure Au and Co. The approach was robust enough to manage the computational challenges due to spin polarization effects introduced by Co atoms and the complexity of real colloidal samples, such as their size distribution and short-range order. Since the DFT + U provided more accurate results than PBE and GLLBSC also for other magnetic-plasmonic (Au-Fe) and plasmonic (Au-Ag) NPs, the proposed approach emerged as the starting choice for the quantitative prediction of the optical properties of plasmonic nanostructures. This is facilitated by the exploitability of already available experimental dielectric functions for pure metals and for some bulk alloys, which are required to identify the best U for each element in the nanoalloys with any composition. Hence, the approach is important for moving beyond the mere identification of the LSP position and width, which are not easily identifiable in all the plasmonic nanoalloys and are not enough to prove the quantitative reliability of the predictions about other plasmonic in the real experimental conditions. In fact, the accuracy reached in modelling and conceptual understanding of innovative plasmonic nanoalloys is a prerequisite for the rational discovery of plasmonic alloys and the quantitative prediction of their advanced properties exploitable in strategic sectors such as quantum optics, magneto-optics, magneto-plasmonics, optical sensing, metamaterials, chiral catalysis and plasmon-enhanced catalysis.

## Methods
### Synthesis
Alloy NPs were obtained by LAL in ethanol (HPLC grade, >99.9%, Sigma-Aldrich) with 1064 nm (6 ns, 50 Hz) pulses of a Nd:YAG laser focused on a metal target with a lens (f 10 cm) at a fluence of 9.5 J cm$^{-2}$. The target with Au:Co 73:27 atomic ratio (>99.9%, Mateck) was placed at the

bottom of a glassy batch chamber mounted on a motorized XY scanning stage (Standa) and kept under an Ar atmosphere. Then, the colloid was concentrated with a rotavapor at 30 °C and mixed 1:1 v/v with an aqueous solution containing ethylenediaminetetraacetic acid disodium salt dihydrate (disodium EDTA, >99%, Sigma-Aldrich, 3 mg mL$^{-1}$) and 0.24 mg mL$^{-1}$ of thiolated polyethylene glycol (PEG-SH, 2000 g mol$^{-1}$, >99%, Laysan Bio). The mixture was kept for 1 h at 60 °C, then washed multiple times with distilled water using dialysis concentration membranes (Sartorius, 10,000 g mol$^{-1}$) until redispersion in distilled water.

The SBS exploits a series of centrifugation steps with increasing centrifugal force as described in Fig. 1c. After centrifugation at 50 × g for 1 h, the supernatant was centrifuged at 300 × g for 1 h. Then the sediment was resuspended in water, centrifuged another time at 300 × g for 1 h and redispersed in water (sample A) after discarding the second supernatant. The supernatant of the first 300 × g run was centrifuged at 750 × g for 1 h. Then the sediment was resuspended in water, centrifuged another time at 750 × g for 1 h and redispersed in water (sample B) after discarding the second supernatant. The supernatant of the first 750 × g run was centrifuged another time at 750 × g for 1 h, and the supernatant was collected (sample C).

Au NPs samples A, B, and C were obtained from a pure Au target through the same synthetic procedure (LAL and SBS) of the Au-Co nanoalloy.

The synthesis of Au-Fe and Au-Ag nanoalloys followed the same synthesis (and characterization) procedure, starting from an Au:Fe 73:27 atomic ratio (>99.9%, from Mateck) or an Au:Ag 40:60 atomic ratio (>99.99%, custom made) target, respectively. In case of the Au-Ag nanoalloys, EDTA was not added to the colloid.

All the NPs samples of this study (name, composition, size) are listed in the Supplementary Table 5.

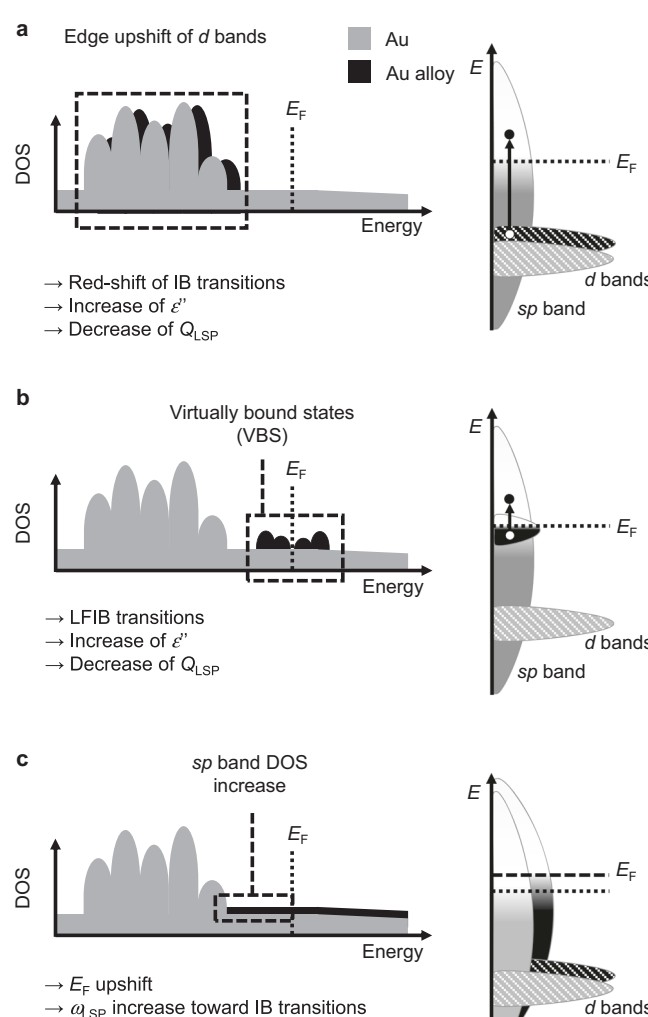

**Fig. 7 | Overview of the three main effects on the LSP of Au when it is alloyed with Co. a** Upshift of the onset of $d$ bands, leading to the red shift of IB transitions and, hence, an increase of $\varepsilon''$ and decrease of $Q_{LSP}$. **b** Appearance of VBS near $E_F$, introducing LFIB and increasing $\varepsilon''$, which means a decrease of $Q_{LSP}$. **c** Increase of electron density in the $sp$ band, leading to the upshift of the $E_F$ and the increase of LSP frequency ($\omega_{LSP}$) towards that of IB transitions, which also implies the decrease of $Q_{LSP}$. Colors in the graphs are: grey (schematic of the DOS of pure Au), black (schematic of the DOS change in the Au-Co alloy).

Size histograms and TEM images of the Au, Au-Fe and Au-Ag samples of this study are reported in Supplementary Note 8.

## Characterization

UV-visible absorption spectroscopy was performed with a JASCO V770 UV−vis−NIR spectrometer in 2 mm quartz cells. Bright-field TEM analysis was performed with a FEI Tecnai G2 12 operating at 100 kV and equipped with a TVIPS CCD camera. Samples were prepared by evaporating the colloids on a copper grid coated with an amorphous carbon holey film. Statistics considered >500 NPs for each sample, using the ImageJ software. HRTEM and EDX analysis was performed with a Talos F200S (Thermofisher Scientific) instrument operating at 200 kV. Elemental maps were obtained from the Au M and Co K lines. ICP-MS measurements were performed with an ICP-MS Agilent 7700 × (Agilent Technologies, equipped with an octupole collision cell operating in kinetic energy discrimination mode). Multistandard IV-ICPMS-71A (Inorganic-Ventures) and Multistandard IMS-103 (Ultra-Scientific) were used for the calibration of all elements. Acid digestion of the NPs samples was done with aqua regia (three parts of HCl 37%

(Sigma Aldrich) and one part of $HNO_3$ 69% w/w "ARISTAR") at 90 °C for 20 min.

XRD analysis was performed with a Panalytical XPert 3 Powder diffractometer equipped with a Cu tube (40 kV, 40 mA), a BBHD mirror, a spinner and a PIXcel detector. The samples were deposed on Si zero-background substrates by drop-casting and drying at room temperature. The diffractograms were analyzed with TOPAS Academic V6 (Bruker AXS) and ICSD databases (005-3764 for Au, 061-1731 and 010-7974 for Au-Co, 005-3764, 040-1295 and 005-6758 for Au-Fe, 060-4769 for Au-Ag). Rietveld refinements were carried out by fitting the background with a Chebyshev function and the required phases. The shape of the reflections was modelled through the fundamental parameter approach incorporated in the program, separating the instrumental and the sample contributions. Fit indicators $R_{wp}$, $R_{exp}$, and GoF (Goodness of Fit) were used to assess the quality of the refined structural models.

For the ellipsometry measurements, the surface of the metallic samples (Au, Ag, Cu, Co and Fe from Sigma-Aldrich, 99.99%; Au(70) Co(30) and Au(73)Fe(27) from Mateck, 99.9%; Au(40)Ag(60) and Au(75)Cu(25) home-made by furnace melting of 99.99% pure single elements) was mirror-polished. The composition and homogeneity of the bulk alloys were assessed by XRD (Rietveld refinement) and EDX, resulting in substitutional solid solutions in all cases. Ellipsometry quantities $\Psi$ and $\Delta$ were measured using a J.A. Woollam V-VASE Spectroscopic Ellipsometer in vertical configuration, at three different angles of incidence (55°, 65°, 75°) in the wavelength range 300-1000 nm. The dielectric functions were evaluated from $\Psi$, $\Delta$ and using WVASE32 ellipsometry data analysis software.

X-ray absorption spectra of samples A, B and C at the Au $L_3$-edge (11,919 eV) were acquired at the XAFS beamline at the Elettra synchrotron radiation facility in Trieste, Italy, using a Si (111) double crystal monochromator and ensuring high-order harmonic rejection by detuning of the second crystal. A water-cooled Pt-coated silicon mirror was used to obtain vertical collimation of the beam. The energy calibration was made by measuring the absorption spectrum of metallic Au foils. For the measurements, the colloidal samples were spotted onto Millipore cellulose filters and vacuum dried at room temperature. The samples were measured in the fluorescence mode at room temperature. X-ray signal extraction and normalization were performed using the ATHENA code, belonging to the set of interactive programs IFEFFIT. The pre-edge background was fitted with a straight line and the post-edge background with a cubic spline. EXAFS data analysis was performed using the EXCURVE code using a $k^2$ weighing scheme.

## DFT calculations

Cells and geometries were optimized using the Quantum-Espresso package[78]. The plane wave basis set was truncated at 35 Ry, the cutoff on density was set to 300 Ry and exchange-correlation effects were included through the PBE functional[79]. The GBRV library[80] of ultrasoft pseudopotential was used. $\Delta n(r)$ was computed as the difference between the electron density of the alloy model and the density of the $2 \times 2 \times 2$ supercell of Au; for $\Delta n(r)$ calculations only, atomic coordinates were kept fixed in FCC positions, while cell parameters were taken from optimized structures. Mixing enthalpies are computed as the difference between the alloy energy and the sum of energies of atomic constituents in their bulk elements. To evaluate Fermi level shifts, electrostatic potentials within cells were aligned taking as reference the potential surrounding an Au atom not having Co as NN, within the procedure described in ref. 54.

Dielectric functions were computed with the GPAW code[81,82] using optimized geometries obtained from Quantum-Espresso and the Hubbard-corrected Local Density Approximation (LDA + U) to the xc functional. Wavefunctions were expanded over a plane-wave basis set with a cutoff of 340 eV. SCF runs were performed using a k-points

density of 15.5 Å and a small Fermi-Dirac smearing of 0.01 eV was imposed. Bands were then refined over a denser grid (density 20 Å) and, finally, dielectric functions were obtained within the IPA with a broadening parameter of 0.1 eV and a Drude scattering rate of 0.01 eV. Details on the U parameterization for Au, Co and the other elements (Fe, Ag, Cu) are reported in Supplementary Note 2.

In all cases, relativistic effects were taken into account through the scalar-field approximation.

### Electrodynamics calculations

Mie model calculations were performed in the 200–1200 nm range with a multipolar order of 5. The NPs' dispersity in size was considered by the experimental size histogram as input for the calculations, to obtain the average extinction cross section for each sample. For each size bin (1 nm), the dielectric function was corrected for the intrinsic size effects as described in refs. 34,57. Briefly, the *A* parameter was set to the standard value of 1, although it can also be optimized for each specific nanoalloy to improve the accuracy of the model[34,57], and the other physical parameters for each element in the alloy were taken from refs. 77,83,84. and averaged by weighting on the composition of each alloy[26,59]. The water solvent refractive index at 25 °C in the 200–1200 nm range was taken from ref. 85.

### Reporting summary

Further information on research design is available in the Nature Portfolio Reporting Summary linked to this article.

## Data availability

The data that support the findings of this study are available from the corresponding authors upon request. Source data are provided with this paper.

## Code availability

All codes used for this study are available from the corresponding authors upon request.

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

## Acknowledgements

L. Avakyan is gratefully acknowledged for help with the Python scripts for DFT + U calculations. This research was funded by the University of Padova P-DiSC project "DYNAMO" (A.V.). We acknowledge the University of Padova Strategic Research Infrastructure Grant 2017: "CAPRI: Calcolo ad Alte Prestazioni per la Ricerca e l'Innovazione" and the CINECA award under the ISCRA initiative, for the availability of high-performance computing resources and support. The Elettra synchrotron radiation facility is acknowledged for provision of beamtime (experiment no. 20210189). M. F. and P.G. acknowledge support from the Ministero dell'Università e della Ricerca (MUR) and the University of Pavia through the program "Dipartimenti di Eccellenza 2023–2027".

## Author contributions

V.A. conceived and supervised the research and acquired funds. V.A. and D.F. designed the research. V.C. performed the research. V.A. and D.F. contributed to modelling. D.B., P.P., P.G. M.F. and A.M. performed additional experiments. V.A., D.F. and V.C. analyzed and interpreted the data. V.C., D.F. and V.A. wrote the manuscript. All authors discussed the data and reviewed the original manuscript.

## Competing interests

The authors declare no competing interests.
