## [Peer Review File · Nature Communications]

Accurate prediction of the optical properties of nanoalloys with both plasmonic and magnetic elementsReviewers' comments:

Reviewer #1 (Remarks to the Author):

To the Authors:

This work presents a combined experimental and computational effort to lay a framework for predicting the optical behavior of Au-Co nanoparticles, based on DFT+U. The work is sounding and potentially of broad interest to the general readership of Nature Commun considering the importance of metallic alloys for photonics and photocatalysis. Thus, I recommend the manuscript for publication if all questions and comments below can be fully addressed. Please find my comments and questions below.

Major issues:

- 1) Overall, the methodology presented in the manuscript could be ultimately validated if tested on Au-Ag and Au-Cu alloy model systems. The dielectric function of some compositions for these alloys have been reported in the literature and would allow for an "apples to apples" direct comparison of how accurate the DFT+U approach is. While the material combination chosen in the work is interesting and certainly has relevant applications, the lack of experimental data as permittivity input leads to the lack of a definitive validation of the modelling approach presented here.
- 2) How was the contribution of Au and Co computed for the calculated permittivity of the alloys? It is known that the contribution of each metal to the dielectric function of an alloy is not weighted by the content of each metal. This critical point needs to be explained in detail in the manuscript.
- 3) A side-by-side comparison should be made between experiments and calculations by measuring the samples via ellipsometry as this technique will provide experimental ϵ' and ϵ'' . I strongly suggest using ellipsometry and comparing the experimental data with the DFT+U predictions as a proof-of-concept. Could the Authors explain why they chose measuring UV-Vis absorption instead?
- 4) How can we be sure that the Au and Co content within the NPs is even (50%-50%) by the EDX maps? A quantitative analysis of the EDX data should be added. And how about the spatial distribution? Can a core-shell-type distribution be excluded?
- 5) Why is the LSP signal for sample B almost quenched? An explanation should be added to the manuscript.
- 6) The author mentioned that this method is independent of the size, size distribution, and stoichiometry of the alloy NPs from Page 10 Line 12 to Page 11 Line 2. However, it is well reported in literature that the sizes of the nanostructure would shift the plasmonic resonances, such as increasing sizes will red-shift the resonance. Comparisons between the experimental (Sample) and simulated model (SRO and SQS) on Figure 4 also shows the blue-shifts, which can be caused by the smaller nanoparticles in the sample. Would adding the considerations of sizes of nanostructure improve the accuracy of the model?
- 7) Did the authors measure the permittivity of pure Co? How does it compare with the calculations and with experimental results previously reported in the literature?

Minor issues:

- 1) All figures should be adjusted to be compliant with the readers' expectations. Some panels are missing labels, the color choice makes it difficult to see the data in some cases. Publications in Nature Commun usually have very informative and polished figures; please review all.
- 2) Figure 1: Color choice for EDS map should be changed (the use of red and green should be avoided because of color blind readers). Figure 1d: legends on the STEM-EDX images are too small to identify.
- 3) Figure 2: Add NP sizes to caption.
- 4) Figure 3 should be improved in quality. Font sizes are small and, as the most relevant figure of the paper, should be presented in a clearer manner. IN panel A: label ϵ' and ϵ'' for both Au and Co, labels are missing.

- 5) Page 5, last lines: what is the actual accuracy of the DFT+U method? Please replace "high" by a quantitative metric.
- 6) I suggest avoiding calling permittivity 'optical constant' as it depends on wavelength. Consider reviewing manuscript for correct terminology.

Reviewer #2 (Remarks to the Author):

The manuscript "Accurate prediction of the optical properties of nanoalloys with plasmonic and magnetic elements" by V. Coviello et al. presents an investigation on the calculation of optical response of AuCo nanoparticles produced by laser ablation in liquid phase. The authors give a detailed description on the procedure for the calculation of the optical properties, consisting on the use of density functional theory to compute the alloy dielectric function, its correction for intrinsic size-effects and the calculation of the extinction efficiency by means of Mie theory and taking into account polydispersity. The computation shows a remarkable similarity with the measured data. The topic is of interest, the fabrication of this specific metastable alloy is valuable and the overall methodology seems correct. However, in my opinion the paper is not suitable for Nature Communications due to the limited novelty of the proposed approach as well as the difficulties to assess the conclusions that are drawn by the authors.

The authors claim their work "paves the way to accurate modelling of plasmonic alloys" but in essence the method (use of DFT calculations for the dielectric function of plasmonic alloys and electrodynamics calculations to compute specific optical response that is compared with experimental data) has been reported in many of the cited works. The authors certainly include some elements related to the specific study, like the choice of the Hubbard correction to avoid issues with other functionals in case of magnetic materials or the size correction of the dielectric function to account for intrinsic size effects. Yet, these are well-known procedures and their incorporation in the plasmonic alloy modeling approach represents an incremental improvement with respect to previous works.

In addition, several claims regarding the predictive power of the approach can not be assessed from the presented results. Although the use of the Hubbard correction may be more appropriate for magnetic materials, it is not clear at what extent is performing better than standard functionals (the comparison in the Suppl. Info if for Au and not for Co). The Drude scattering rate is ad-hoc fixed to 0.01 eV regardless composition, same as the size-correction parameter use to account for size effects, that is equal to 1 for all materials although the Fermi velocity for Au and Co can be quite different and it is questionable that the approximation can be useful for very small (6 nm) particles. These are standard values used in literature, but indicate that the overall procedure contains several approximations. In this context, concluding that the approach allows insight on the alloy internal structure (with no other experimental confirmation) because the short range ordered structures, in average, agree better than the random structure is not well grounded. The comparison with linear averaging of the dielectric function, a procedure that is known to be inappropriate for alloys - as the authors mention, has no relevance. In my opinion, the novelty claims should be significantly reduced and the comparison should justify that the choice of the presented approach (for instance, comparing the extinction efficiency with results obtained using standard DFT functionals, employing no size-corrected dielectric function and computing the dielectric function with other choices of the Drude scattering rate). Such study could be certainly valuable for a more specialized journal.

Reviewer #3 (Remarks to the Author):

In this article the authors use a combination of DFT in the LDA+U approximation for the exchange correlation potential, along with corrections for intrinsic size effects and Mie theory to simulate extinction spectra for Au-Co alloy nanoparticles.

The work presented is an interesting study of the impact of various factors on the simulation of dielectric function of these alloys. However, I don't feel that it represents an example of significantly new or noteworthy results. It is well understood that the addition of metals with partially filled d-bands (such as Co) will strongly damp plasmonic responses, as is observed here. Similarly it is also well understood that the details of the crystal structures of a supercell, when attempting to model a disordered alloy, will depend on the atomic arrangements. It has also been well-established that the LDA+U approach can improve the simulation of dielectric functions for appropriate materials. The sample preparation methodology appears to offer some advantages over other nanoparticle synthesis techniques, but the authors have previously published about the method elsewhere.

The agreement between the simulations and experimental data is not compelling. But this is typical for DFT-IPA simulations of plasmonic materials. Furthermore, given the large number of variables involved in the simulations (intrinsic size effects, U parameter, crystal structures) there are potentially many different combinations that could equally well achieve the same result. I don't feel that the authors have presented a reproducible approach that could be applied more universally beyond this particular system.

A few more specific points:

- The justification that there is a residual plasmon peak in the Au-Co spectra if figure 2d is not convincingly made. It could equally well be argued that the features remaining are purely due to interband transitions. The dielectric functions shown in Figure S4 indicate that plasmons would only be found well outside the visible range.
- The justification that $U=2$ is the optimum for Au is not presented.
- The agreement between experiment and calculation for Co is very poor. Selecting the anti-ferromagnetic result when that doesn't correspond to the ground state for Co is not appropriate. It is more likely that the chosen approach to simulate the dielectric function is not correct.
- Are there any parameters applied in the intrinsic size effects and, if so, how were these determined.

Finally, although generally well written and clear, there are numerous minor grammatical mistakes throughout the manuscript. I recommend the authors seek further feedback on the writing before any possible resubmission.

Reviewer #4 (Remarks to the Author):

In the present manuscript, the authors report the colloidal synthesis via laser ablation followed by size sorting and computational modeling of the optical spectrum of AuCo nanolloys. Samples with 3 different average sizes and Co content were collected and characterized. From the modeling side, DFT+U with U parametrized on the pure metals was used to predict the Au-Co dielectric function and hence the spectral response via the Mie model. Two types of chemical ordering were investigated at the DFT+U level, i.e., ordered (SRO) and random (SQS) configurations. The experimentally observed quenching of the plasmon for AuCo alloys is well reproduced by theory, especially by the SRO model rather than the SQS model, however after adjusting the cell parameter to experiment.

A detailed analysis shows that three main phenomena, i.e., the upshift of the d-band edge, the appearance of virtually bound state and the increase in conduction band electron density, are singled out and rationalized, which in my opinion is the most original part of the present manuscript.

Given the interest of the field and the good comparison between theory and experiment hereby achieved, I think that the present manuscript can be accepted for publication on Nature Communications, after the following two remarks are addressed.

Chemical ordering can be ordered or random but can also be segregation, which is actually likely in

the AuCo system, although perhaps not under the present synthesis conditions. I remind the authors however that the Vegard's law basically works also for segregated (e.g. core-shell) nanoalloys.

The sentence "Unfortunately, the in silico prediction of nanoalloys properties is seldom complete and satisfactory and, in fact, the accurate comparison with experimental results beyond a qualitative agreement is not demonstrated for metals containing not-noble elements." is surprising. For small or medium-sized clusters, such as especially but not exclusively ligand-protected, there are in fact a number of papers which demonstrate that it is possible to predict quantitatively the photo-absorption spectrum of both monometallic and bimetallic systems, including systems mixing a plasmonic metal with a non-noble metal, see Refs. [DOI: 10.1063/5.0056869; DOI: 10.1021/acs.jpcc.9b09060; DOI: 10.1021/acs.jpcc.8b00556; DOI: 10.1021/acsnano.0c01183; DOI: 10.1021/acs.jpcc.9b07382; DOI: 10.1021/acs.jpcclett.8b00308; DOI: 10.1021/jacs.0c05685; DOI: 10.1021/jp508824w]. For other systems, see also Refs.[DOI: 10.1039/c8cp04107e; DOI: 10.1063/1.4996687; DOI: 10.1021/acs.jpcc.6b04709; DOI: 10.1021/jp2087219; DOI:10.1021/jp112217g]. These studies demonstrate that a predictive computational approach to the modeling of the photo-absorption spectrum of complex nanoalloys do exist and can realistically reach simulations of particles of 4 nm in size. Clearly, it rests to be seen whether this approach is feasible or is necessary, and whether more simplified schemes such as investigated here can afford a similar accuracy at a much smaller computational expense, but it cannot be stated that such approaches do not exist. Incidentally, the procedure followed by the authors of adjusting the cell parameter to calculate the dielectric function which witnesses a pronounced sensitivity of the optical response to geometry is not different conceptually from what has been demonstrated in [DOI: 10.1063/5.0056869] and earlier papers, i.e., use experimental X-ray geometries to achieve predictive modeling of the optical response of small (binary) clusters.

Dear Editor,

Please find enclosed the revised version of the manuscript entitled
“Accurate prediction of the optical properties of nanoalloys with plasmonic and magnetic elements”
Article NCOMMS-23-06386A-Z

The manuscript has been revised based on all the Reviewer’s suggestions and by responding to all their comments, also with the intensive implementation of the experimental results where expressly requested by the Referees or appropriate for providing the best and clearest answers.

In this regard, we would like to thank all the Referees for their appreciation of our work and useful comments, which have been helpful to improve the quality of the manuscript and provide additional corroboration to its contents, general scope and novelty.

Yours sincerely,
V. Amendola, on behalf of all Authors

Reviewer #1

This work presents a combined experimental and computational effort to lay a framework for predicting the optical behavior of Au-Co nanoparticles, based on DFT+U. The work is sounding and potentially of broad interest to the general readership of Nature Commun considering the importance of metallic alloys for photonics and photocatalysis. Thus, I recommend the manuscript for publication if all questions and comments below can be fully addressed. Please find my comments and questions below.

We thank the reviewer for the general appreciation of our work and useful comments.

Major issues:

1) Overall, the methodology presented in the manuscript could be ultimately validated if tested on Au-Ag and Au-Cu alloy model systems. The dielectric function of some compositions for these alloys have been reported in the literature and would allow for an “apples to apples” direct comparison of how accurate the DFT+U approach is. While the material combination chosen in the work is interesting and certainly has relevant applications, the lack of experimental data as permittivity input leads to the lack of a definitive validation of the modelling approach presented here.

The comparison of DFT+U calculations with the experimental and calculated dielectric functions reported in literature for an Au-Ag and an Au-Cu bulk alloy was added to the revised manuscript. Besides, we added the comparison for another magnetic-plasmonic material (Au-Fe) for which the dielectric function is available in literature.

Finally, to provide validation of the generality of our method, the predictions of the optical properties of real colloidal samples of nanoparticles made of another magnetic-plasmonic alloy (Au-Fe) and a typical plasmonic alloy (Au-Ag) were also added.

The revised manuscript now reads:

“Besides, its general applicability was successfully verified with another magnetic-plasmonic nanoalloy (Au-Fe) and a typical plasmonic nanoalloy (Au-Ag).”

And

“The general applicability of the method was also substantiated with another magnetic-plasmonic nanoalloy (Au-Fe) and a typical plasmonic nanoalloy (Au-Ag), as well as by comparison with experimental and calculated bulk dielectric functions.”

And

“The general applicability of the method was further validated with other magnetic-plasmonic and standard plasmonic alloys. First, the predictive ability was tested against the dielectric functions reported in literature for some Au-Fe, Au-Ag and Au-Cu alloys, as well as those experimentally measured in this study (Section S6 in S.I.). To keep the approach equal to the Au-Co case and as general as possible, the procedure started with the identification of the appropriate U parameters from the dielectric function of the pure elements, which are always available in literature with good accuracy (Section S2 in S.I.). In principle, U could be best obtained

from the alloy dielectric functions, when these are available and accurate. However, it is worth noticing that ellipsometry measurements are not trivial and are known to lead to different results depending on a series of parameters, principally related to the experimental set-up (in vacuum, less frequent and more expensive, or in air, which is easier but subjected to oxidation and contamination of the metal surface), and to the structure and composition of the bulk samples (homogeneity, roughness, purity, chemical transformations).^{17,43,44} This is apparent from the variable agreement between our measurements and the literature values for alloys (Figure S10 in S.I.), even for single elements (Figures S4 in S.I.). Nonetheless, the dielectric functions calculated with the DFT+U for the magnetic-plasmonic and the plasmonic alloys always exhibited a fair agreement with the literature values in the whole spectral range from 200 to 1000 nm (Figure S10 in S.I.).

As it is the focus of this study, the optical properties of real nanoalloy colloidal samples of another magnetic-plasmonic system (Au-Fe) and of a typical plasmonic system (Au-Ag) were considered (Figure 5). Also in this case, the quality of our method is evident from the R^2 , which are 0.96129 for Au(80)Fe(20) NPs sample A, 0.96978 R^2 for Au(83)Fe(17) NPs sample B; 0.97738 for Au(45)Ag(55) NPs sample A and 0.93222 for Au(45)Ag(55) NPs sample B. Note that standard calculation parameters (A term = 1, Drude scattering rate = 0.01 eV) and generic SQS cells were used, without any optimization for each specific sample, which could lead to higher R^2 . Despite this, the dielectric functions calculated with the DFT+U allowed the prediction of the optical properties of real nanoalloys with higher accuracy than the literature permittivity from experiments^{45,66} and DFT calculations.¹⁷”

And

“**Figure 5.** Comparison of experimental and calculated mass extinction coefficients (in $\text{mL mg}^{-1} \text{cm}^{-1}$) for colloidal samples of nanoalloys made of another magnetic-plasmonic system (Au-Fe) and of a typical plasmonic system (Au-Ag). The calculations obtained with the dielectric functions available in literature from experimental measurements^{45,66} and DFT calculations¹⁷ are also reported. The agreement with the real samples of nanoalloys is evaluated quantitatively with the R^2 .”

And

“The general applicability of the method was substantiated also with another magnetic-plasmonic nanoalloy (Au-Fe) and a typical plasmonic nanoalloy (Au-Ag), as well as by comparison with experimental and calculated bulk dielectric functions.”

And the new section in S.I. entitled

“**S6. Comparison of calculated and experimental dielectric functions of bulk alloys**”

2) How was the contribution of Au and Co computed for the calculated permittivity of the alloys? It is known that the contribution of each metal to the dielectric function of an alloy is not weighted by the content of each metal. This critical point needs to be explained in detail in the manuscript.

This point was clarified in the revised manuscript, which reads:

“Since no experimental dielectric functions are available in literature for the Au-Co alloys, **instead of resorting to inaccurate methods such as the weight of single metal dielectric functions**, the DFT+U method coupled with IPA was used to calculate these functions.^{40,41} The overall procedure (Figure 3a) starts with the parametrization of U for Au 5d and Co 3d states versus the experimental dielectric function of pure Au and Co reported in literature (see Section S2 in S.I.).^{43,60} Then, the same values of U were transferred to Au-Co alloys **with compositions equivalent to the experimental samples**. This is legitimate because U changes with the element oxidation state but it remains almost unchanged when going from pure metals to their alloys.⁴¹”

3) A side-by-side comparison should be made between experiments and calculations by measuring the samples via ellipsometry as this technique will provide experimental ϵ' and ϵ'' . I strongly suggest using ellipsometry and comparing the experimental data with the DFT+U predictions as a proof-of-concept.

The comparison of DFT+U predictions and experimentally measured dielectric functions for the Au-Co alloy, as well as for the Au-Fe, Au-Ag and Au-Cu alloys, were added to the revised manuscript, and the difficulty of obtaining the same experimental results from independent ellipsometry experiments was also discussed (see also the answer to the first and next points).

The revised manuscript now reads:

“The general applicability of the method was further validated with other magnetic-plasmonic and standard plasmonic alloys. First, the predictive ability was tested against the dielectric functions reported in literature for some Au-Fe, Au-Ag and Au-Cu alloys, as well as those experimentally measured in this study (Section S6

in S.I.). To keep the approach equal to the Au-Co case and as general as possible, the procedure started with the identification of the appropriate U parameters from the dielectric function of the pure elements, which are always available in literature with good accuracy (Section S2 in S.I.). In principle, U could be best obtained from the alloy dielectric functions, when these are available and accurate. However, it is worth noticing that ellipsometry measurements are not trivial and are known to lead to different results depending on a series of parameters, principally related to the experimental set-up (in vacuum, less frequent and more expensive, or in air, which is easier but subjected to oxidation and contamination of the metal surface), and to the structure and composition of the bulk samples (homogeneity, roughness, purity, chemical transformations).^{17,43,44} This is apparent from the variable agreement between our measurements and the literature values for alloys (Figure S10 in S.I.), even for single elements (Figures S4 in S.I.). Nonetheless, the dielectric functions calculated with the DFT+U for the magnetic-plasmonic and the plasmonic alloys always exhibited a fair agreement with the literature values in the whole spectral range from 200 to 1000 nm (Figure S10 in S.I.).”

And the new section in S.I. entitled

“S6. Comparison of calculated and experimental dielectric functions of bulk alloys”

Could the Authors explain why they chose measuring UV-Vis absorption instead?

The novelty and importance represented by the modelling of optical extinction, instead of limiting to the dielectric function from ellipsometry, was clarified in the revised manuscript, which reads:

“Even for pure noble metals, the experimental measurement of the dielectric function by ellipsometry is difficult, due to its dependence on film density, roughness, purity and measurement range, **leading to different results in literature.**^{17,43,44} This task becomes very challenging in the case of multielement metals subject to oxidation or forming metastable alloys prone to segregation.^{17,18,45} **Indeed, the advantage of using computed dielectric functions in place of experimental ones is that of overcoming both the experimental challenges related to their measurement and the thermodynamic constraints that hinder the access to optical properties of metastable materials.**”

And

“The μ_{ext} is representative of the plasmonic properties of nanoscale objects, which are the focus of this study. The experimental UV-vis spectra of a colloid of NPs collected with different optical absorption spectrometers are identical within an experimental error of a few %, and the mass of NPs in the sample can be measured accurately by ICP-MS using certified standards and procedures, therefore μ_{ext} is an accurate parameter for benchmarking the predictions of our model. Conversely, the dielectric functions measured by ellipsometry may change from one laboratory to another due to a series of experimental factors, especially for immiscible elements but even for pure noble metals like Au.^{43,44}”

And

“The effective ability of the calculated dielectric functions to reproduce the optical properties of real samples can be assessed only by computing the μ_{ext} of the Au-Co samples with the Mie model using either the SRO- (Figure 4a) or the SQS (Figure 4b) cells and comparing them to the μ_{ext} quantified by ICP-MS.”

And

“However, it is worth noticing that ellipsometry measurements are not trivial and are known to lead to different results depending on a series of parameters, principally related to the experimental set-up (in vacuum, less frequent and more expensive, or in air, which is easier but subjected to oxidation and contamination of the metal surface), and to the structure and composition of the bulk samples (homogeneity, roughness, purity, chemical transformations).^{17,43,44} This is apparent from the variable agreement between our measurements and the literature values for alloys (Figure S10 in S.I.), even for single elements (Figures S4 in S.I.).”

And

“As it is the focus of this study, the optical properties of real nanoalloy colloidal samples of another magnetic-plasmonic system (Au-Fe) and of a typical plasmonic system (Au-Ag) were considered (Figure 5).”

4) How can we be sure that the Au and Co content within the NPs is even (50%-50%) by the EDX maps? A quantitative analysis of the EDX data should be added. And how about the spatial distribution? Can a core-shell-type distribution be excluded?

We added additional experimental results to clarify these points in the revised manuscript, which now reads:

“The use of Vegard’s law for alloys is justified because STEM-EDX allowed excluding the formation of core-shell structures (see Figure 1d and Section S1 in S.I.), which is possible in NPs of immiscible elements.^{50,58,59}”

And

“The average composition was confirmed further by additional STEM-EDX quantitative analysis performed on several single NPs of each sample (Table S1 in S.I.).”

And the new section in S.I. entitled

“**S1. Additional STEM-EDX analysis on Au-Co nanoalloys**”

5) *Why is the LSP signal for sample B almost quenched? An explanation should be added to the manuscript.*

We modified the text to avoid misinterpretation at this point in the revised manuscript, which reads:

“Sample A exhibits a broad band in the 475 – 500 nm range, which could be attributed to a blue-shifted LSP or to interband transitions. The LSP is not well identifiable in the Au-Co B sample, while a weak plasmon absorption centered at 505 nm is observed in the Au-Co C sample (i.e. the sample richest in Au).”

and

“The effect is the significant decrease of the Q_{LSP} in Au-Co alloys, corresponding to weak or unidentifiable LSP in samples A, B and C in Figure 2d.”

Besides, the specific interpretation of the plasmonic response of Au-Co nanoalloys is reported in the “*Discussion*” section.

6) *The author mentioned that this method is independent of the size, size distribution, and stoichiometry of the alloy NPs from Page 10 Line 12 to Page 11 Line 2. However, it is well reported in literature that the sizes of the nanostructure would shift the plasmonic resonances, such as increasing sizes will red-shift the resonance. Comparisons between the experimental (Sample) and simulated model (SRO and SQS) on Figure 4 also shows the blue-shifts, which can be caused by the smaller nanoparticles in the sample. Would adding the considerations of sizes of nanostructure improve the accuracy of the model?*

We modified the text to clarify that the experimentally measured size of the nanoalloys was included in our calculations, as indicated by the Referee. Hence, the spectral shift of the calculated spectra and of the experimental ones are the result of both the composition and the size distribution of each sample.

The revised manuscript now reads:

“The dielectric functions were used in the Mie model to calculate the mass extinction coefficient (μ_{ext}). Since the optical properties of nanoparticles are size dependent, the accuracy of the Mie model was maximised by using the TEM-measured size (d_i) histogram of each Au-Co NPs sample as input (i.e. the Mie calculation was performed for each size bin of the size histogram and then weighted for the relative frequency). Besides, at each size bin in the TEM histogram, the extrinsic size effects were accounted for by setting the multipolarity order to 4 and the dielectric function was corrected for the intrinsic size effects with the well-established procedure proposed by Kreibig and based on the Drude-Lorentz model.^{34,56}”

And

“The R^2 indicates the good agreement between the SRO- calculations and the experimental curves in all three samples, meaning that the procedure worked well despite the differences in the average size, size distribution and stoichiometry of the alloy NPs.”

7) *Did the authors measure the permittivity of pure Co? How does it compare with the calculations and with experimental results previously reported in the literature?*

We added the requested experimental and literature results to clarify these points in the revised manuscript and remark that the calculations for pure metals are used only to identify a suitable U parameter for the calculation of the dielectric function in the alloys, because the dielectric function of pure elements is already available experimentally, unlike the majority of alloys.

The new section in S.I. entitled

“**S2. Parameterization of the Hubbard term and comparison with the experimental dielectric functions of pure metals**”

Reads

“The dielectric functions of elemental Au, Co, Fe, Ag and Cu computed with the LDA+U are compared also with experimental curves from ellipsometry measurements performed in this work (Figure S4).”

And

“Figure S4. Comparison of the dielectric functions of elemental Au, Co, Fe, Ag and Cu computed with the LDA+U with experimental curves from literature and from ellipsometry measurements performed in this work.”

And the revised manuscript reads

“The value of U must be optimized, preferably against available experimental data, such as the dielectric function of elemental constituents of the alloy, to match the targeted properties of the system.”

And

“To keep the approach equal to the Au-Co case and as general as possible, the procedure started with the identification of the appropriate U parameters from the dielectric function of the pure elements, which are always available in literature with good accuracy (Section S2 in S.I.). In principle, U could be best obtained from the alloy dielectric functions, when these are available and accurate.”

Minor issues:

1) All figures should be adjusted to be compliant with the readers' expectations. Some panels are missing labels, the color choice makes it difficult to see the data in some cases. Publications in Nature Commun usually have very informative and polished figures; please review all.

The figures in the revised manuscript were modified as suggested at this point and the following ones.

2) Figure 1: Color choice for EDS map should be changed (the use of red and green should be avoided because of color blind readers). Figure 1d: legends on the STEM-EDX images are too small to identify.

The EDX figures in the revised manuscript were modified as suggested.

3) Figure 2: Add NP sizes to caption.

This information was added to the caption of Figure 2.

4) Figure 3 should be improved in quality. Font sizes are small and, as the most relevant figure of the paper, should be presented in a clearer manner. IN panel A: label ϵ' and ϵ'' for both Au and Co, labels are missing.

Figure 3 in the revised manuscript was modified as suggested.

5) Page 5, last lines: what is the actual accuracy of the DFT+U method? Please replace “high” by a quantitative metric.

In the revised manuscript we introduced the coefficient of determination (R^2) to compare quantitatively the accuracy of calculations for different samples.

The revised manuscript now reads:

“The integrated computational-experimental endeavour reproduced the optical properties of realistic systems such as the colloidal solutions of Au-Co plasmonic nanoalloys with different composition and size distribution, with coefficients of determination (R^2) larger than 0.97.”

And

“The coefficient of determination (Figure 4c) was used to quantify the agreement between calculations and experiments among different samples and models.”

6) I suggest avoiding calling permittivity ‘optical constant’ as it depends on wavelength. Consider reviewing manuscript for correct terminology.

We thank the Reviewer for highlighting this issue, which was fixed in the revised manuscript.

Reviewer #2

The manuscript “Accurate prediction of the optical properties of nanoalloys with plasmonic and magnetic elements” by V. Coviello et al. presents an investigation on the calculation of optical response of AuCo

nanoparticles produced by laser ablation in liquid phase. The authors give a detailed description on the procedure for the calculation of the optical properties, consisting on the use of density functional theory to compute the alloy dielectric function, its correction for intrinsic size-effects and the calculation of the extinction efficiency by means of Mie theory and taking into account polydispersity. The computation shows a remarkable similarity with the measured data. The topic is of interest, the fabrication of this specific metastable alloy is valuable and the overall methodology seems correct. However, in my opinion the paper is not suitable for Nature Communications due to the limited novelty of the proposed approach as well as the difficulties to assess the conclusions that are drawn by the authors.

The authors claim their work “paves the way to accurate modelling of plasmonic alloys” but in essence the method (use of DFT calculations for the dielectric function of plasmonic alloys and electrodynamic calculations to compute specific optical response that is compared with experimental data) has been reported in many of the cited works. The authors certainly include some elements related to the specific study, like the choice of the Hubbard correction to avoid issues with other functionals in case of magnetic materials or the size correction of the dielectric function to account for intrinsic size effects. Yet, these are well-known procedures and their incorporation in the plasmonic alloy modeling approach represents an incremental improvement with respect to previous works.

We thank the reviewer for the general appreciation of our work and useful comments.

The novelty of our study resides in the prediction of extinction spectra of real samples of magnetic-plasmonic alloy nanoparticles using calculated dielectric functions in place of experimental ones, within an approach that was never validated before for alloys containing plasmonic and magnetic elements. We also show that computed dielectric functions can be better references than experimental ones, which are difficult to measure and unavailable for most materials. We believe that this represents a remarkable step forward over previous achievements, providing unprecedented support for the mindful synthesis of new plasmonic nanoalloys. The novelty of our study has now been clarified in the revised manuscript

- For what concern the significance of our work

“This work shows that the modelling of the optical properties of nanoalloys can now meet the accuracy level of the experiment, thus paving the way to the computational investigation of plasmonic alloys still waiting for assessment and exploitation in fundamental sectors such as quantum optics, magneto-optics, magneto-plasmonics, metamaterials, chiral catalysis and plasmon-enhanced catalysis.”

And

“Indeed, the advantage of using computed dielectric functions in place of experimental ones is that of overcoming both the experimental challenges related to their measurement and the thermodynamic constraints that hinder the access to optical properties of metastable materials.”

And

“The general applicability of the method was also substantiated with another magnetic-plasmonic nanoalloy (Au-Fe) and a typical plasmonic nanoalloy (Au-Ag), as well as by comparison with experimental and calculated bulk dielectric functions. Hence, this study shows that the modelling of the optical properties of nanoalloys can now meet the accuracy level of the experiment, thus paving the way for the mindful development of innovative plasmonic nanoalloys with tailored properties.”

And

“The general applicability of the method was substantiated also with another magnetic-plasmonic nanoalloy (Au-Fe) and a typical plasmonic nanoalloy (Au-Ag), as well as by comparison with experimental and calculated bulk dielectric functions. Hence, this study advanced the level of accuracy for synthesis, modelling and conceptual understanding of innovative plasmonic nanoalloys.”

- For what concerns the advance represented by the accurate reproduction of the mass extinction coefficient of *real samples of magnetic-plasmonic nanoalloys in the form of a colloid* instead of limiting the study to the dielectric function, which has not been reported previously in literature

“Unfortunately, the *in silico* prediction of nanoalloys properties is limited to metal clusters and the accurate comparison with experimental results beyond a qualitative agreement is not demonstrated to date in the size range above just a few nanometers.”

And

“For instance, the dielectric functions of plasmonic alloys of coinage and noble metals, which have full *d* levels and are non-magnetic, were recently calculated with linear-response time-dependent DFT and quantitatively

compared with experimental results in terms of plasmon peak position and width, while no information was provided about the accuracy of the absolute extinction coefficient in real samples of plasmonic nanoalloys.^{17,37}

And

“Nevertheless, the DFT+U method was not yet applied to the quantitative prediction and conceptual understanding of the optical properties of real samples of alloy NPs.”

And

“The effective ability of the calculated dielectric functions to reproduce the optical properties of real samples can be assessed only by computing the μ_{ext} of the Au-Co samples with the Mie model using either the SRO- (Figure 4a) or the SQS (Figure 4b) cells and comparing them to the μ_{ext} quantified by ICP-MS.”

And

“The general applicability of the method was further validated with other magnetic-plasmonic and standard plasmonic alloys.”

And

“As it is the focus of this study, the optical properties of real nanoalloy colloidal samples of another magnetic-plasmonic system (Au-Fe) and of a typical plasmonic system (Au-Ag) were considered (Figure 5).”

And

“the dielectric functions calculated with the DFT+U allowed the prediction of the optical properties of real nanoalloys with higher accuracy than the literature permittivity from experiments^{45,66} and DFT calculations.¹⁷”

- For what concerns the analysis of the several parameters (supercell structure, DFT functional type, Drude broadening factor, size correction, A parameter) involved in the quantitative prediction of the optical properties of *real samples of magnetic-plasmonic nanoalloys in the form of a colloid*, which has been introduced to answer the Referees' requests and has no equivalent in literature for this type of samples and for the calculation of the mass extinction coefficient,

“After assessing that the short-range order is relevant for the accuracy of the calculations, also the effect of the functional type (Figure 4c and Section S5 in S.I.), the Drude scattering rate, the correction for intrinsic size effects and the A parameter for the size correction were quantified (Figure 4c). The results confirm that the LDA+U functional is advantageous for the accuracy of the calculations, while the Drude scattering rate and the A parameter only affect the 4th or 3rd decimal place of the R^2 , thus the standard values adopted in literature^{17,26,34,41,56} are the appropriate choice in absence of information guiding to the optimization for each specific nanoalloy. The correction for intrinsic size effect accounts for a change on the 3rd decimal place of the R^2 in samples A and B, where the plasmon resonance is significantly damped and the NPs size is larger, but it is on the 2nd decimal place of the R^2 in sample C, where the plasmon resonance is more intense because of the lower amount of Co in the alloy and the smaller size of the NPs. Overall, the analysis shows that the accuracy of the calculations with our method was not the result of a casual combination of these parameters.”

- For what concerns the potential of inferring information about the short-range order in *real samples of magnetic-plasmonic nanoalloys in the form of a colloid*, based on the quantitative agreement between the calculated and experimental mass extinction coefficient, which has no equivalent in literature for this type of samples and has been demonstrated with additional experiments in the revised manuscript,

“Consequently, the role of the chemical order in the physical and chemical behaviour of nanoalloys is still not well understood and the possibility to study its effect through accurate computational approaches would be a major advance in the field.”

And

“In this way, the quantitative comparison of the calculated and experimental optical properties of the nanoalloys provided insight on their “ultrastructure”, namely the structural features not directly identifiable with standard characterization approaches such as electron microscopy. We are not aware of other examples of inferring the existence of short-range order in real samples of nanoalloys from the accuracy of DFT calculations of their optical properties. This advance in the synergy between models and experiments is crucial for guiding further investigations with complex and expensive equipment which is not easily accessible. These findings provide also further evidence to the previous theoretical indications that the SRO in nanoalloys has a measurable effect on the optical properties, and modelling efforts should be tailored to the actual atomic arrangement of the experimental system under examination.^{17,41}”

- For what concerns the interpretation of the plasmonic response of Au-Co nanoalloys from the in-depth analysis of their electronic structure, which was not reported before for this magnetic-plasmonic alloy in literature

“Hence, the Au-Co nanoalloy is of enormous interest for quantum plasmonics, magneto-plasmonics, chiral catalysis, responsive optical nanomaterials, sensing and, in general, for benchmarking the research capabilities

in this field.^{15,18,29} However, little is known about the optical properties of this magnetic-plasmonic nanomaterial, also because the Au-Co solid solution is metastable which, together with the diverse oxidation potentials of the two metals, imply the maximum synthetic challenges for a nanoalloy.^{29,50–52}

In addition, several claims regarding the predictive power of the approach can not be assessed from the presented results. Although the use of the Hubbard correction may be more appropriate for magnetic materials, it is not clear at what extent is performing better than standard functionals (the comparison in the Suppl. Info if for Au and not for Co).

The predictive power of the approach compared to the standard potentials PBE was clarified in the revised manuscript, which reads:

“After assessing that the short-range order is relevant for the accuracy of the calculations, also the effect of the functional type (Figure 4c and Section S5 in S.I.), the Drude scattering rate, the correction for intrinsic size effects and the A parameter for the size correction were quantified (Figure 4c). The results confirm that the LDA+U functional is advantageous for the accuracy of the calculations, while the Drude scattering rate and the A parameter only affect the 4th or 3rd decimal place of the R^2 , thus the standard values adopted in literature^{17,26,34,41,56} are the appropriate choice in absence of information guiding to the optimization for each specific nanoalloy.”

And the new section in S.I. entitled

“S5. Comparison of the mass extinction coefficients of Au-Co nanoalloys calculated with the LDA+U and PBE functionals“

Which reads

“**Figure S9.** Comparison of mass extinction coefficients calculated with the LDA+U and PBE functionals for the Au-Co samples (left: SQS cells; right: SRO cells). The arrows indicate the region of the plasmon resonance in the nanoalloys, where the PBE calculations resulted in lower accuracy than the LDA+U functional, as expected from ref.⁴ and in agreement with the results obtained with pure Au (see Figure S2). Note that the numerical stability of more recent functionals such as the GLLBSC one does not support a plasmonic alloy with magnetic elements,^{5–8} and were not considered in this study.”

Besides, in the revised manuscript we have clarified that the calculation of pure element dielectric function is not the scope of this study because these functions are already available in literature. Instead, the calculations for pure elements are exploited to identify a suitable U parameter for that element in the calculation of the dielectric function of the alloy. Where the dielectric function of an alloy is experimentally available with enough accuracy, the U parameter can be obtained directly from the alloy.

The revised manuscript now reads:

“To keep the approach equal to the Au-Co case and as general as possible, the procedure started with the identification of the appropriate U parameters from the dielectric function of the pure elements, which are always available in literature with good accuracy (Section S2 in S.I.). In principle, U could be best obtained from the alloy dielectric functions, when these are available and accurate.”

The Drude scattering rate is ad-hoc fixed to 0.01 eV regardless composition, same as the size-correction parameter use to account for size effects, that is equal to 1 for all materials although the Fermi velocity for Au and Co can be quite different and it is questionable that the approximation can be useful for very small (6 nm) particles. These are standard values used in literature, but indicate that the overall procedure contains several approximations.

In the revised manuscript we have demonstrated quantitatively that the values for Drude scattering rate and the A parameter have a minor or negligible effect on the results of the calculations for the magnetic-plasmonic Au-Co NPs and that standard values based on literature are an appropriate choice in the absence of specific information.

The revised manuscript reads:

“After assessing that the short-range order is relevant for the accuracy of the calculations, also the effect of the functional type (Figure 4c and Section S5 in S.I.), the Drude scattering rate, the correction for intrinsic size effects and the A parameter for the size correction were quantified (Figure 4c). The results confirm that the LDA+U functional is advantageous for the accuracy of the calculations, while the Drude scattering rate and the A parameter only affect the 4th or 3rd decimal place of the R^2 , thus the standard values adopted in literature^{17,26,34,41,56} are the appropriate choice in absence of information guiding to the optimization for each

specific nanoalloy. The correction for intrinsic size effect accounts for a change on the 3rd decimal place of the R^2 in samples A and B, where the plasmon resonance is significantly damped and the NPs size is larger, but it is on the 2nd decimal place of the R^2 in sample C, where the plasmon resonance is more intense because of the lower amount of Co in the alloy and the smaller size of the NPs. Overall, the analysis shows that the accuracy of the calculations with our method was not the result of a casual combination of these parameters.”

And, in caption of Figure 4

“(c) R^2 quantifying the accuracy of calculated data to experimental ones for the various parameters and models considered in the study (for details see text).”

In this context, concluding that the approach allows insight on the alloy internal structure (with no other experimental confirmation) because the short range ordered structures, in average, agree better than the random structure is not well grounded.

Additional experimental evidence about the short-range order of the nanoalloys in Sample A was provided by measuring the extended X-ray absorption fine structure (EXAFS) with synchrotron light source.

The revised manuscript now reads:

“However, the R^2 of the SRO- model for sample A is lower than in samples B and C, resulting in a comparable value with the SQS model. Indeed, the Au-Co sample A has the largest spread between the Co content estimated from the lattice parameter (19 at%) and that obtained from the ICP-MS analysis (29 at%), suggesting a positive deviation from the Vegard’s law which is typically observed in alloys with short-range elements segregation.^{64,65} This may be indicative of the faster cooling rate of smaller NPs generated during LAL, which has been shown to favour homogeneous mixing instead of phase segregation in immiscible alloys.^{58,59} To gather more experimental confirmation about the short-range order of the samples, the extended X-ray absorption fine structure (EXAFS) at the Au L_3 -edge was measured and analysed (see Section S3 in S.I.). Since this measurement provides a specific insight into the structural environment of Au atoms, it was possible to calculate the interatomic distance of Au nearest neighbour shell from the fitting of the EXAFS signal. The results (summarized in Figure S7 in S.I.) clearly show a deviation of the gold atoms interatomic distance in sample A, compared to samples B and C, versus the experimentally measured content of Co, which substantiate the short-order segregation in the nanoalloy. Hence, a further model of the Au(24)Co(8) was adopted, which contemplates a short-range Co segregation arrangement (SRO+, see Figure 4a and Section S4 in S.I.), i.e. a positive WC parameter. As shown by the plot of the μ_{ext} and R^2 (Figure 4b-c), the SQS+ model is the best at reproducing the experimental result for sample A.”

And the new section in S.I. entitled

“**S3. EXAFS analysis on Au-Co nanoalloys**”

The comparison with linear averaging of the dielectric function, a procedure that is known to be inappropriate for alloys - as the authors mention, has no relevance. In my opinion, the novelty claims should be significantly reduced and the comparison should justify that the choice of the presented approach (for instance, comparing the extinction efficiency with results obtained using standard DFT functionals, employing no size-corrected dielectric function and computing the dielectric function with other choices of the Drude scattering rate). Such study could be certainly valuable for a more specialized journal.

We added the comparison with the results from the standard functional PBE and the other parameters mentioned by the Referee (see the answer to previous points) in the revised manuscript, which now reads:

“After assessing that the short-range order is relevant for the accuracy of the calculations, also the effect of the functional type (Figure 4c and Section S5 in S.I.), the Drude scattering rate, the correction for intrinsic size effects and the A parameter for the size correction were quantified (Figure 4c). The results confirm that the LDA+U functional is advantageous for the accuracy of the calculations, while the Drude scattering rate and the A parameter only affect the 4th or 3rd decimal place of the R^2 , thus the standard values adopted in literature^{17,26,34,41,56} are the appropriate choice in absence of information guiding to the optimization for each specific nanoalloy. The correction for intrinsic size effect accounts for a change on the 3rd decimal place of the R^2 in samples A and B, where the plasmon resonance is significantly damped and the NPs size is larger, but it is on the 2nd decimal place of the R^2 in sample C, where the plasmon resonance is more intense because of the lower amount of Co in the alloy and the smaller size of the NPs. Overall, the analysis shows that the accuracy of the calculations with our method was not the result of a casual combination of these parameters.

The general applicability of the method was further validated with other magnetic-plasmonic and standard plasmonic alloys. First, the predictive ability was tested against the dielectric functions reported in literature for some Au-Fe, Au-Ag and Au-Cu alloys, as well as those experimentally measured in this study (Section S6 in S.I.). To keep the approach equal to the Au-Co case and as general as possible, the procedure started with the identification of the appropriate U parameters from the dielectric function of the pure elements, which are always available in literature with good accuracy (Section S2 in S.I.). In principle, U could be best obtained from the alloy dielectric functions, when these are available and accurate. However, it is worth noticing that ellipsometry measurements are not trivial and are known to lead to different results depending on a series of parameters, principally related to the experimental set-up (in vacuum, less frequent and more expensive, or in air, which is easier but subjected to oxidation and contamination of the metal surface), and to the structure and composition of the bulk samples (homogeneity, roughness, purity, chemical transformations).^{17,43,44} This is apparent from the variable agreement between our measurements and the literature values for alloys (Figure S10 in S.I.), even for single elements (Figures S4 in S.I.). Nonetheless, the dielectric functions calculated with the DFT+U for the magnetic-plasmonic and the plasmonic alloys always exhibited a fair agreement with the literature values in the whole spectral range from 200 to 1000 nm (Figure S10 in S.I.).

As it is the focus of this study, the optical properties of real nanoalloy colloidal samples of another magnetic-plasmonic system (Au-Fe) and of a typical plasmonic system (Au-Ag) were considered (Figure 5). Also in this case, the quality of our method is evident from the R^2 , which are 0.96129 for Au(80)Fe(20) NPs sample A, 0.96978 R^2 for Au(83)Fe(17) NPs sample B; 0.97738 for Au(45)Ag(55) NPs sample A and 0.93222 for Au(45)Ag(55) NPs sample B. Note that standard calculation parameters (A term = 1, Drude scattering rate = 0.01 eV) and generic SQS cells were used, without any optimization for each specific sample, which could lead to higher R^2 . Despite this, the dielectric functions calculated with the DFT+U allowed the prediction of the optical properties of real nanoalloys with higher accuracy than the literature permittivity from experiments^{45,66} and DFT calculations.¹⁷

And, in caption of Figure 4

“(c) R^2 quantifying the accuracy of calculated data to experimental ones for the various parameters and models considered in the study (for details see text).”

And, in caption of the new Figure 5

“**Figure 5.** Comparison of experimental and calculated mass extinction coefficients (in $\text{mL mg}^{-1} \text{cm}^{-1}$) for colloidal samples of nanoalloys made of another magnetic-plasmonic system (Au-Fe) and of a typical plasmonic system (Au-Ag). The calculations obtained with the dielectric functions available in literature from experimental measurements^{45,66} and DFT calculations¹⁷ are also reported. The agreement with the real samples of nanoalloys is evaluated quantitatively with the R^2 .”

And the new sections in S.I. entitled

“**S5. Comparison of the mass extinction coefficients of Au-Co nanoalloys calculated with the LDA+U and PBE functionals**”

And

“**S6. Comparison of calculated and experimental dielectric functions of bulk alloys**”

Concerning the advances provided by this manuscript to the field, which is the accurate prediction of the optical properties of *real samples of magnetic-plasmonic nanoalloys and plasmonic nanoalloys*, we have revised the manuscript to clarify this aspect (see the answer to the first point).

About this last point, we want to stress the challenge of obtaining quantitative agreement with real colloidal samples of metastable nanoalloys of magnetic and plasmonic elements, as well as of plasmonic nanoalloys, for which there are no other examples in literature. This challenge is well represented by the comparison of the optical properties of the real samples of Au-Fe and Au-Ag nanoalloys with those obtained with the experimental or calculated dielectric functions available in literature, which performed worse than our method (new Figure 5 in the revised manuscript).

Reviewer #3

In this article the authors use a combination of DFT in the LDA+U approximation for the exchange correlation potential, along with corrections for intrinsic size effects and Mie theory to simulate extinction spectra for Au-Co alloy nanoparticles.

The work presented is an interesting study of the impact of various factors on the simulation of dielectric function of these alloys. However, I don't feel that it represents an example of significantly new or noteworthy results. It is well understood that the addition of metals with partially filled d-bands (such as Co) will strongly damp plasmonic responses, as is observed here. Similarly it is also well understood that the details of the crystal structures of a supercell, when attempting to model a disordered alloy, will depend on the atomic arrangements. It has also been well-established that the LDA+U approach can improve the simulation of dielectric functions for appropriate materials. The sample preparation methodology appears to offer some advantages over other nanoparticle synthesis techniques, but the authors have previously published about the method elsewhere.

We thank the reviewer for the general appreciation of our work and useful comments.

The arguments of the Referee are related to the comparison of the calculated and experimental dielectric functions of nanoalloys, or to the comparison of plasmon peak position and full-width half maximum with the optical properties of real nanoalloys.

In this study, instead, we have tackled the challenge of the quantitative prediction of the optical properties of real samples of magnetic-plasmonic nanoalloys and plasmonic nanoalloys, for which there are no other examples in literature. This challenge is well represented by the comparison of the optical properties of the real samples of Au-Fe and Au-Ag nanoalloys with those obtained with the experimental or calculated dielectric functions available in literature, which performed worse than the method adopted in the present work (new Figure 5 in the revised manuscript).

Hence, the novelty of our study has been clarified in the revised manuscript, which reads:

- For what concern the significance of our work

“This work shows that the modelling of the optical properties of nanoalloys can now meet the accuracy level of the experiment, thus paving the way to the computational investigation of plasmonic alloys still waiting for assessment and exploitation in fundamental sectors such as quantum optics, magneto-optics, magneto-plasmonics, metamaterials, chiral catalysis and plasmon-enhanced catalysis.”

And

“Indeed, the advantage of using computed dielectric functions in place of experimental ones is that of overcoming both the experimental challenges related to their measurement and the thermodynamic constraints that hinder the access to optical properties of metastable materials.”

And

“The general applicability of the method was also substantiated with another magnetic-plasmonic nanoalloy (Au-Fe) and a typical plasmonic nanoalloy (Au-Ag), as well as by comparison with experimental and calculated bulk dielectric functions. Hence, this study shows that the modelling of the optical properties of nanoalloys can now meet the accuracy level of the experiment, thus paving the way for the mindful development of innovative plasmonic nanoalloys with tailored properties.”

And

“The general applicability of the method was substantiated also with another magnetic-plasmonic nanoalloy (Au-Fe) and a typical plasmonic nanoalloy (Au-Ag), as well as by comparison with experimental and calculated bulk dielectric functions. Hence, this study advanced the level of accuracy for synthesis, modelling and conceptual understanding of innovative plasmonic nanoalloys.”

- For what concerns the advance represented by the accurate reproduction of the mass extinction coefficient of *real samples of magnetic-plasmonic nanoalloys in the form of a colloid* instead of limiting the study to the dielectric function, which has not been reported previously in literature

“Unfortunately, the *in silico* prediction of nanoalloys properties is limited to metal clusters and the accurate comparison with experimental results beyond a qualitative agreement is not demonstrated to date in the size range above just a few nanometers.”

And

“For instance, the dielectric functions of plasmonic alloys of coinage and noble metals, which have full *d* levels and are non-magnetic, were recently calculated with linear-response time-dependent DFT and quantitatively compared with experimental results in terms of plasmon peak position and width, while no information was provided about the accuracy of the absolute extinction coefficient in real samples of plasmonic nanoalloys.^{17,37}”

And

“Nevertheless, the DFT+U method was not yet applied to the quantitative prediction and conceptual understanding of the optical properties of real samples of alloy NPs.”

And

“The effective ability of the calculated dielectric functions to reproduce the optical properties of real samples can be assessed only by computing the μ_{ext} of the Au-Co samples with the Mie model using either the SRO- (Figure 4a) or the SQS (Figure 4b) cells and comparing them to the μ_{ext} quantified by ICP-MS.”

And

“The general applicability of the method was further validated with other magnetic-plasmonic and standard plasmonic alloys.”

And

“As it is the focus of this study, the optical properties of real nanoalloy colloidal samples of another magnetic-plasmonic system (Au-Fe) and of a typical plasmonic system (Au-Ag) were considered (Figure 5).”

And

“the dielectric functions calculated with the DFT+U allowed the prediction of the optical properties of real nanoalloys with higher accuracy than the literature permittivity from experiments^{45,66} and DFT calculations.¹⁷”

- For what concerns the analysis of the several parameters (supercell structure, DFT functional type, Drude broadening factor, size correction, A parameter) involved in the quantitative prediction of the optical properties of *real samples of magnetic-plasmonic nanoalloys in the form of a colloid*, which has been introduced to answer the Referees' requests and has no equivalent in literature for this type of samples and for the calculation of the mass extinction coefficient,

“After assessing that the short-range order is relevant for the accuracy of the calculations, also the effect of the functional type (Figure 4c and Section S5 in S.I.), the Drude scattering rate, the correction for intrinsic size effects and the A parameter for the size correction were quantified (Figure 4c). The results confirm that the LDA+U functional is advantageous for the accuracy of the calculations, while the Drude scattering rate and the A parameter only affect the 4th or 3rd decimal place of the R^2 , thus the standard values adopted in literature^{17,26,34,41,56} are the appropriate choice in absence of information guiding to the optimization for each specific nanoalloy. The correction for intrinsic size effect accounts for a change on the 3rd decimal place of the R^2 in samples A and B, where the plasmon resonance is significantly damped and the NPs size is larger, but it is on the 2nd decimal place of the R^2 in sample C, where the plasmon resonance is more intense because of the lower amount of Co in the alloy and the smaller size of the NPs. Overall, the analysis shows that the accuracy of the calculations with our method was not the result of a casual combination of these parameters.”

- For what concerns the potential of inferring information about the short-range order in *real samples of magnetic-plasmonic nanoalloys in the form of a colloid*, based on the quantitative agreement between the calculated and experimental mass extinction coefficient, which has no equivalent in literature for this type of samples and has been demonstrated with additional experiments in the revised manuscript,

“Consequently, the role of the chemical order in the physical and chemical behaviour of nanoalloys is still not well understood and the possibility to study its effect through accurate computational approaches would be a major advance in the field.”

And

“In this way, the quantitative comparison of the calculated and experimental optical properties of the nanoalloys provided insight on their “ultrastructure”, namely the structural features not directly identifiable with standard characterization approaches such as electron microscopy. We are not aware of other examples of inferring the existence of short-range order in real samples of nanoalloys from the accuracy of DFT calculations of their optical properties. This advance in the synergy between models and experiments is crucial for guiding further investigations with complex and expensive equipment which is not easily accessible. These findings provide also further evidence to the previous theoretical indications that the SRO in nanoalloys has a measurable effect on the optical properties, and modelling efforts should be tailored to the actual atomic arrangement of the experimental system under examination.^{17,41}”

- For what concerns the interpretation of the plasmonic response of Au-Co nanoalloys from the in-depth analysis of their electronic structure, which was not reported before for this magnetic-plasmonic alloy in literature

“Hence, the Au-Co nanoalloy is of enormous interest for quantum plasmonics, magneto-plasmonics, chiral catalysis, responsive optical nanomaterials, sensing and, in general, for benchmarking the research capabilities in this field.^{15,18,29} However, little is known about the optical properties of this magnetic-plasmonic nanomaterial, also because the Au-Co solid solution is metastable which, together with the diverse oxidation potentials of the two metals, imply the maximum synthetic challenges for a nanoalloy.^{29,50–52}”

- Concerning the role of the synthesis method for this study, we have not claimed the novelty of the synthesis method but we have just highlighted that it is a state of the art method for accessing alloys of

plasmonic and magnetic elements which are immiscible and not achievable as colloid through other approaches. This technique was key to accessing the nanomaterials for benchmarking the modelling machinery and, during the revisions of the manuscript, its versatility allowed access to colloids of another magnetic-plasmonic nanoalloy (Au-Fe) and to a typical plasmonic nanoalloy (Au-Ag). This is stated in the manuscripts, which reads:

“In parallel, the synthetic methodologies relied on state-of-the-art laser synthesis and sorting of metastable alloy nanoparticles. The approach was robust enough to face the computational challenges due to spin polarization effects introduced by Co atoms, as well as the experimental difficulties due to the immiscibility of the Au-Co system”

The agreement between the simulations and experimental data is not compelling. But this is typical for DFT-IPA simulations of plasmonic materials.

As discussed in the answer to the previous point, we are not aware of other examples in literature where DFT-IPA calculations were used to predict quantitatively the mass extinction coefficient in real samples of magnetic-plasmonic nanoalloys.

The goodness of the mass extinction coefficients calculated for real samples of magnetic-plasmonic nanoalloys was clarified in the revised manuscript, which reads:

“Nevertheless, the DFT+U method was not yet applied to the **quantitative prediction** and conceptual understanding of the optical properties of real samples of alloy NPs.”

And

“Due to the lack of other studies about the **quantitative agreement between calculated and experimental extinction coefficients in real samples of magnetic-plasmonic or plasmonic nanoalloys with sizes of several nm**, the goodness of R^2 (0.97140, 0.98571 and 0.98166 for Au-Co samples A, B and C) can be appreciated by comparison with the values of 0.94707 (Au A), 0.98718 (Au B) and 0.98324 (Au C), which were achieved for the experimental samples of Au NPs shown in Figure 2d, using one of the best experimental gold dielectric functions⁴³ and applying the Mie model to the TEM measured size histograms of the Au NPs (as done for the Au-Co NPs).”

And

“the dielectric functions calculated with the DFT+U allowed the prediction of the optical properties of real nanoalloys with higher accuracy than the literature permittivity from experiments^{45,66} and DFT calculations.¹⁷”

Furthermore, given the large number of variables involved in the simulations (intrinsic size effects, U parameter, crystal structures) there are potentially many different combinations that could equally well achieve the same result. I don't feel that the authors have presented a reproducible approach that could be applied more universally beyond this particular system.

The analysis of the parameters (cell structure, DFT functional type, Drude broadening factor, size correction, A parameter) involved in the quantitative prediction of the optical properties of *real samples of magnetic-plasmonic nanoalloys in the form of a colloid*, which has no equivalent in literature for this type of samples and for the calculation of the mass extinction coefficient, was added to the revised manuscript:

“After assessing that the short-range order is relevant for the accuracy of the calculations, also the effect of the functional type (Figure 4c and Section S5 in S.I.), the Drude scattering rate, the correction for intrinsic size effects and the A parameter for the size correction were quantified (Figure 4c). The results confirm that the LDA+U functional is advantageous for the accuracy of the calculations, while the Drude scattering rate and the A parameter only affect the 4th or 3rd decimal place of the R^2 , thus the standard values adopted in literature^{17,26,34,41,56} are the appropriate choice in absence of information guiding to the optimization for each specific nanoalloy. The correction for intrinsic size effect accounts for a change on the 3rd decimal place of the R^2 in samples A and B, where the plasmon resonance is significantly damped and the NPs size is larger, but it is on the 2nd decimal place of the R^2 in sample C, where the plasmon resonance is more intense because of the lower amount of Co in the alloy and the smaller size of the NPs. Overall, the analysis shows that the accuracy of the calculations with our method was not the result of a casual combination of these parameters.”

Besides the reproducibility of the approach has been demonstrated with NPs of another type of magnetic-plasmonic nanoalloy (Au-Fe) and of a typical plasmonic nanoalloy (AuAg).

The revised manuscript now reads:

“As it is the focus of this study, the optical properties of real nanoalloy colloidal samples of another magnetic-plasmonic system (Au-Fe) and of a typical plasmonic system (Au-Ag) were considered (Figure 5). Also in this case, the quality of our method is evident from the R^2 , which are 0.96129 for Au(80)Fe(20) NPs sample A, 0.96978 R^2 for Au(83)Fe(17) NPs sample B; 0.97738 for Au(45)Ag(55) NPs sample A and 0.93222 for Au(45)Ag(55) NPs sample B. Note that standard calculation parameters (A term = 1, Drude scattering rate = 0.01 eV) and generic SQS cells were used, without any optimization for each specific sample, which could lead to higher R^2 . Despite this, the dielectric functions calculated with the DFT+U allowed the prediction of the optical properties of real nanoalloys with higher accuracy than the literature permittivity from experiments^{45,66} and DFT calculations.¹⁷”

And

“**Figure 5.** Comparison of experimental and calculated mass extinction coefficients (in mL mg⁻¹ cm⁻¹) for colloidal samples of nanoalloys made of another magnetic-plasmonic system (Au-Fe) and of a typical plasmonic system (Au-Ag). The calculations obtained with the dielectric functions available in literature from experimental measurements^{45,66} and DFT calculations¹⁷ are also reported. The agreement with the real samples of nanoalloys is evaluated quantitatively with the R^2 .”

A few more specific points:

- *The justification that there is a residual plasmon peak in the Au-Co spectra if figure 2d is not convincingly made. it could equally well be argued that the features remaining are purely due to interband transitions. The dielectric functions shown in Figure S4 indicate that plasmons would only be found well outside the visible range.*

This point was clarified in the revised manuscript, which reads:

“Sample A exhibits a broad band in the 475 – 500 nm range, which could be attributed to a blue-shifted LSP or to interband transitions. The LSP is not well identifiable in the Au-Co B sample, while a weak plasmon absorption centered at 505 nm is observed in the Au-Co C sample (i.e. the sample richest in Au).”

And

“In case of the alloy with the composition of sample C, which has an LSP in proximity of 500 nm (Figure 2d), the Frölich condition is also satisfied in the same spectral range (Figure 3d), confirming the plasmonic nature of the weak band in that sample. The same band is not found in sample B (Figure 2d), richer in Co, and indeed the minimum at the Frölich condition is barely detectable.”

- *The justification that $U=2$ is the optimum for Au is not presented.*

This information was added to the revised manuscript (new Figure S3 in S.I.).

- *The agreement between experiment and calculation for Co is very poor. Selecting the anti-ferromagnetic result when that doesn't correspond to the ground state for Co is not appropriate. It is more likely that the chosen approach to simulate the dielectric function is not correct.*

The antiferromagnetic model was dropped and the parameterization of U was done on ferromagnetic Co, following the procedure used for all other elements: The procedure shows that $U = 2$ eV provides the best match. Though the comparison with the experimental spectrum is less satisfactory than that obtained for the other elements, it still allows the identification of the best value of U, that is the purpose of the analysis on the pure element.

This point was clarified in the revised manuscript, which reads

“To keep the approach equal to the Au-Co case and as general as possible, the procedure started with the identification of the appropriate U parameters from the dielectric function of the pure elements, which are always available in literature with good accuracy (Section S2 in S.I.). In principle, U could be best obtained from the alloy dielectric functions, when these are available and accurate. However, it is worth noticing that ellipsometry measurements are not trivial and are known to lead to different results depending on a series of parameters, principally related to the experimental set-up (in vacuum, less frequent and more expensive, or in air, which is easier but subjected to oxidation and contamination of the metal surface), and to the structure and composition of the bulk samples (homogeneity, roughness, purity, chemical transformations).^{17,43,44} This is apparent from the variable agreement between our measurements and the literature values for alloys (Figure S10 in S.I.), even for single elements (Figures S4 in S.I.).”

And the sections in S.I. entitled

“S2. Parameterization of the Hubbard term and comparison with the experimental dielectric functions of pure metals“

And

“Figure S3. Calculated dielectric functions for selected values of the U parameter. PBE results are also shown.”

And

“Table S2. R² values of the comparison between calculated and experimental dielectric functions for selected values of the Hubbard parameter U. The best matches are highlighted in green. U = 0 corresponds to pure LDA results. PBE results are also reported.”

And the new sections in S.I. entitled

“S6. Comparison of calculated and experimental dielectric functions of bulk alloys“

Reads

“The R² are reported in Table S4. Note that the result for the Au-Co alloy (R² of 0.980 and 0.886 for ε' and ε'’, respectively) provides further confirmation that the choice of U = 2 for Co is appropriate. Hence, although the optimization of U on the dielectric function of a bulk alloy with a composition similar to the NPs is easier, the use of the pure element dielectric functions also leads to appropriate results.”

- Are there any parameters applied in the intrinsic size effects and, if so, how were these determined.

This information was added to the revised manuscript, which reads:

“For each size bin (1 nm), the dielectric function was corrected for the intrinsic size effects as described in ref.^{34,56}. Briefly, the A parameter was set to the standard value of 1, although it can also be optimized for each specific nanoalloy to improve the accuracy of the model,^{34,56} and the other physical parameters for each element in the alloy were taken from ref.⁸²⁻⁸⁴ and averaged by weighting on the composition of each alloy, following what previously described in ref.^{26,58”}

Finally, although generally well written and clear, there are numerous minor grammatical mistakes throughout the manuscript. I recommend the authors seek further feedback on the writing before any possible resubmission.

The manuscript has been checked for grammatical mistakes before resubmission.

Reviewer #4

In the present manuscript, the authors report the colloidal synthesis via laser ablation followed by size sorting and computational modeling of the optical spectrum of AuCo nanolloys. Samples with 3 different average sizes and Co content were collected and characterized. From the modeling side, DFT+U with U parametrized on the pure metals was used to predict the Au-Co dielectric function and hence the spectral response via the Mie model. Two types of chemical ordering were investigated at the DFT+U level, i.e., ordered (SRO) and random (SQS) configurations. The experimentally observed quenching of the plasmon for AuCo alloys is well reproduced by theory, especially by the SRO model rather than the SQS model, however after adjusting the cell parameter to experiment.

A detailed analysis shows that three main phenomena, i.e., the upshift of the d-band edge, the appearance of virtually bound state and the increase in conduction band electron density, are singled out and rationalized, which in my opinion is the most original part of the present manuscript.

Given the interest of the field and the good comparison between theory and experiment hereby achieved, I think that the present manuscript can be accepted for publication on Nature Communications, after the following two remarks are addressed.

We thank the reviewer for the general appreciation of our work and useful comments.

Chemical ordering can be ordered or random but can also be segregation, which is actually likely in the AuCo system, although perhaps not under the present synthesis conditions. I remind the authors however that Vegard's law basically works also for segregated (e.g. core-shell) nanoalloys.

We clarified this point and that the NPs are not core-shell in the revised manuscript, which reads:

“The use of Vegard’s law for alloys is justified because STEM-EDX allowed excluding the formation of core-shell structures (see Figure 1d and Section S1 in S.I.), which is possible in NPs of immiscible elements.^{50,58,59”}

And

“suggesting a positive deviation from the Vegard’s law which is typically observed in alloys with short-range elements segregation.^{64,65”}

And the new sections in S.I. entitled

“ **S1. Additional STEM-EDX analysis on Au-Co nanoalloys**“

Reads

“**Figure S1. Additional STEM-EDX maps on Au-Co NPs (Au M line, Co K line) showing that the NPs do not have a core-shell structure and are homogeneous alloys. The linescan profile of Au and Co in a group of nanoparticles of sample A is also shown** “

The sentence "Unfortunately, the in silico prediction of nanoalloys properties is seldom complete and satisfactory and, in fact, the accurate comparison with experimental results beyond a qualitative agreement is not demonstrated for metals containing not-noble elements." is surprising. For small or medium-sized clusters, such as especially but not exclusively ligand-protected, there are in fact a number of papers which demonstrate that it is possible to predict quantitatively the photo-absorption spectrum of both monometallic and bimetallic systems, including systems mixing a plasmonic metal with a non-noble metal, see Refs. [DOI: 10.1063/5.0056869; DOI: 10.1021/acs.jpcc.9b09060; DOI: 10.1021/acs.jpcc.8b00556; DOI: 10.1021/acsnano.0c01183; DOI: 10.1021/acs.jpcc.9b07382; DOI: 10.1021/acs.jpcc.8b00308; DOI: 10.1021/jacs.0c05685; DOI: 10.1021/jp508824w]. For other systems, see also Refs.[DOI: 10.1039/c8cp04107e; DOI: 10.1063/1.4996687; DOI: 10.1021/acs.jpcc.6b04709; DOI: 10.1021/jp2087219; DOI: 10.1021/jp112217g]. These studies demonstrate that a predictive computational approach to the modeling of the photo-absorption spectrum of complex nanoalloys do exist and can realistically reach simulations of particles of 4 nm in size. Clearly, it rests to be seen whether this approach is feasible or is necessary, and whether more simplified schemes such as investigated here can afford a similar accuracy at a much smaller computational expense, but it cannot be stated that such approaches do not exist.

We clarified this point in the revised manuscript, which reads:

“Unfortunately, the *in silico* prediction of nanoalloys properties is limited to metal clusters and the accurate comparison with experimental results beyond a qualitative agreement is not demonstrated to date in the size range above just a few nanometers.”

And

“Innovative plasmonic nanoalloys are structurally complex, difficult to synthesize and lack a well-identified modelling methodology which is computationally feasible also for NPs with sizes of several or tens of nm.^{67,68}”

Incidentally, the procedure followed by the authors of adjusting the cell parameter to calculate the dielectric function which witnesses a pronounced sensitivity of the optical response to geometry is not different conceptually from what has been demonstrated in [DOI: 10.1063/5.0056869] and earlier papers, i.e., use experimental X-ray geometries to achieve predictive modeling of the optical response of small (binary) clusters.

No ad-hoc adjustments were made in the present work. Only in the case $\Delta n(\mathbf{r})$ calculations, atomic positions were constrained to FCC sites. This has been clarified in the revised manuscript:

“for $\Delta n(\mathbf{r})$ calculations only, atomic coordinates were kept fixed in FCC positions, while cell parameters were taken from optimized structures.”

All the modifications in the revised manuscript have been marked as red text.

Reviewers' comments:

Reviewer #2 (Remarks to the Author):

In the revised version of the manuscript "Accurate prediction of the optical properties of nanoalloys with plasmonic and magnetic elements", the authors provide new experimental data and analysis in order to better support their conclusions. Although the authors have addressed the issues regarding the difficulties in assessing their conclusions, the prediction methodology (computations combining DFT and electrostatics) has limited novelty and it is still my opinion that the paper would be more suitable for a specialized journal.

In any case, although the revision certainly strengthens the paper, the modifications open new issues and several aspects have to be addressed.

- The authors claim a novel approach, stating that prior studies mainly compared plasmon peak characteristics without considering the absolute extinction coefficient. While this may be true, the distinction between these comparisons is largely technical once optical properties are computed. The novelty would be more significant if the authors could demonstrate that analyzing the entire extinction spectra provides additional insights compared to focusing solely on peak characteristics.

- In the revised manuscript, the authors argue that DFT+U yields accuracy and flexibility. However, they point out that it relies on knowing the optimal Hubbard correction, at least for the pure materials in the alloy, which contradicts the claim that using computed data helps overcome experimental challenges.

- However, the main issue with the choice of the functional is that, although it works well for Au, provides a quite inaccurate description for Co properties (new Figures S3 or S4), predicting a peak in the imaginary part of the dielectric function that is absent in the experiment. Then it may be questioned whether and why a poor description of pure materials should be able to provide a very good description of alloys. An intuitive explanation is that the inaccurate description of the Co properties is not too relevant because the alloys are Au-rich and Co plays a lesser role in the overall alloy properties. If this is the case, the advantage of DFT+U with respect to other functionals (like GLLBSC) for tackling magnetic materials seems not relevant. In this line, it should be noted that the choice of PBE, although providing worse dielectric function prediction, also leads to quite high R values and enables to deduce the same conclusions regarding the prevalence of SRO vs SQS structures. Overall, in light of the new data provided, it seems that the choice of DFT+U is not as critical as the authors seem to claim.

- In the revised version the authors introduce the coefficient of determination R to judge the goodness of the description between the model and experimental data. This is a good estimator for model evaluation, but it is questionable how useful is in order to discriminate among models that provide R values close to 1. Namely, when comparing SQS and SRO (or even when comparing DFT+U and PBE) all values seem to be above 0.95. A larger value means better agreement, but it is questionable if the difference is relevant keeping in mind that the experimental error is being ignored in the statistical analysis. Namely, the extinction measurements -as any other measurement- have experimental error (the authors mention "a few percent" in the text), and if the difference between models prediction and experiment is within the experimental error, one can not choose a model in favour of another. Moreover, the alloy composition and particle size, used as input parameters for the simulations have some experimental uncertainty. These aspects might be less critical than ignoring the experimental errors in the extinction measurements (it would be difficult to simulate a cell with the exact composition and the particle size error probably is not important because the whole size distribution is taken into account), but it should also be justified why these experimental uncertainties are ignored when evaluating the goodness of the model. In any case, a relevant judgement of the superiority of one model over another requires taking into the experimental error in measurements.

- Regarding experimental error, the authors claim that the dielectric function measured by ellipsometry may change from one laboratory to another due to a series of experimental factors. The discrepancies among dielectric functions reported in the literature (like in the papers cited by the authors) are not related to ellipsometry but to the differences between samples. Thus, is inappropriate -and unfair- to state that the extinction coefficient measurements are reliable and repeatable (implicitly assuming that the same colloid is being measured) while ellipsometry is not (implicitly assuming that different samples are being measured).

- The authors include the correction for intrinsic size effects as a part of the method. In the revised version of the manuscript, they show that the size-correction parameter (A) and the Drude broadening values do not significantly affect the results. Accordingly, the authors conclude that "The analysis show that the accuracy of the calculations with our method was not the result of a causal combination of these parameters". However, the conclusion that must be extracted from this analysis is that these parameters are irrelevant -at least in the present case-, probably because they basically affect the long-wavelength range of the extinction spectra. This also opens the question of whether taking into account the actual size dispersion is a necessary step because the particles are very small (especially in samples B and C) and might be that neither intrinsic nor extrinsic size effects are relevant, at least in this case.

As a final comment, I sincerely acknowledge the effort of the authors and consider that this a really nice work with a very detailed comparison of optical measurements and sample structure and composition and that the paper is certainly suitable for a very good journal (in my opinion, a more specialized journal). In this sense, I think many of the statements on novelty, uniqueness and reach of the proposed method are questionable, unnecessary and mask the actual value of the work. Thus, the proposal of using DFT and electrodynamics to simulate the optical properties of alloys is well reported in the literature and the choice of DFT+U might be advantageous in this case but it should not be regarded as a general recipe. Taking into account size dispersion and intrinsic size correction might be certainly important, but it does not seem relevant in the present case. The results indicate a better correlation of the simulations with a specific degree of order in the structure but do not allow to state that the method allows inferring the existence of short-range. Overall, I kindly request that the authors review their statements regarding the novelty and significance of their study with a more measured and less speculative tone. Doing so will not diminish the importance of their work but will provide a more accurate representation of its true value.

Reviewer #3 (Remarks to the Author):

The authors have comprehensively addressed the technical concerns raised by the reviewers and the additional data added strengthens the paper.

The work is interesting, scientifically sound and worthy of publication in a scientific journal. However, rather than representing a substantial and novel advance in the field this work is an incremental step. Using calculated dielectric functions to simulate nanoparticle spectra is not novel. The DFT computation methods used are not novel. The approaches to simulating an alloy are not novel. It is a nice demonstration that careful application of well-established techniques can give good agreement with experimental spectra. I am happy to leave it to the editors whether they see such an outcome as being suitable for publication in this journal.

Reviewer #4 (Remarks to the Author):

The authors have revised their manuscript satisfactorily. I recommend publication on Nature Communications.